# Non-Receptor Tyrosine Kinases: Their Structure and Mechanistic Role in Tumor Progression and Resistance

**DOI:** 10.3390/cancers16152754

**Published:** 2024-08-02

**Authors:** Abdulaziz M. Eshaq, Thomas W. Flanagan, Sofie-Yasmin Hassan, Sara A. Al Asheikh, Waleed A. Al-Amoudi, Simeon Santourlidis, Sarah-Lilly Hassan, Maryam O. Alamodi, Marcelo L. Bendhack, Mohammed O. Alamodi, Youssef Haikel, Mossad Megahed, Mohamed Hassan

**Affiliations:** 1Department of Epidemiology and Biostatistics, Milken Institute School of Public Health, George Washington University, Washington, DC 20052, USA; eshaq@gwu.edu; 2College of Medicine, Alfaisal University, Riyadh 11533, Saudi Arabia; saralsheikh@alfaisal.edu (S.A.A.A.); walamoudi@alfaisal.edu (W.A.A.-A.); maryam.o.alamodi@gmail.com (M.O.A.); m.alamodi1998@gmail.com (M.O.A.); 3Department of Pharmacology and Experimental Therapeutics, LSU Health Sciences Center, New Orleans, LA 70112, USA; tflan1@lsuhsc.edu; 4Department of Pharmacy, Faculty of Science, Heinrich-Heine University Duesseldorf, 40225 Duesseldorf, Germany; sofie00@gmx.de; 5Institute of Cell Therapeutics and Diagnostics, University Medical Center of Duesseldorf, 40225 Duesseldorf, Germany; simeon.santourlidis@med.uni-duesseldorf.de; 6Department of Chemistry, Faculty of Science, Heinrich-Heine University Duesseldorf, 40225 Duesseldorf, Germany; saha109@hhu.de; 7Department of Urology, Red Cross University Hospital, Positivo University, Rua Mauá 1111, Curitiba 80030-200, Brazil; marcelo@uro-onco.net; 8Institut National de la Santé et de la Recherche Médicale, University of Strasbourg, 67000 Strasbourg, France; youssef.haikel@unistra.fr; 9Department of Operative Dentistry and Endodontics, Dental Faculty, 67000 Strasbourg, France; 10Pôle de Médecine et Chirurgie Bucco-Dentaire, Hôpital Civil, Hôpitaux Universitaire de Strasbourg, 67000 Strasbourg, France; 11Clinic of Dermatology, University Hospital of Aachen, 52074 Aachen, Germany; mmegahed@ukaachen.de; 12Research Laboratory of Surgery-Oncology, Department of Surgery, Tulane University School of Medicine, New Orleans, LA 70112, USA

**Keywords:** PTK, NRTKs, Ack, Jak, Fes, Fak, Tec, Src, Csk, Abl, Syk kinases

## Abstract

**Simple Summary:**

Protein tyrosine kinases (PTKs) are classified into two groups: one group includes tyrosine kinases, and the second group includes serine/threonine kinases. The group of tyrosine kinases includes both receptor tyrosine kinases (RTKs) and non-receptor tyrosine kinases (NRTKs) that function as “on” or “off” switches for many cellular functions. NRTKs are kinase enzymes which are overexpressed and activated in many cancer types and regulate variable cellular functions, including cell growth and progression and their dependent mechanisms and the associated signaling pathways. Thus, targeting NRTKs is of great interest to improve the treatment strategy of different tumor types.

**Abstract:**

Protein tyrosine kinases (PTKs) function as key molecules in the signaling pathways in addition to their impact as a therapeutic target for the treatment of many human diseases, including cancer. PTKs are characterized by their ability to phosphorylate serine, threonine, or tyrosine residues and can thereby rapidly and reversibly alter the function of their protein substrates in the form of significant changes in protein confirmation and affinity for their interaction with protein partners to drive cellular functions under normal and pathological conditions. PTKs are classified into two groups: one of which represents tyrosine kinases, while the other one includes the members of the serine/threonine kinases. The group of tyrosine kinases is subdivided into subgroups: one of them includes the member of receptor tyrosine kinases (RTKs), while the other subgroup includes the member of non-receptor tyrosine kinases (NRTKs). Both these kinase groups function as an “on” or "off" switch in many cellular functions. NRTKs are enzymes which are overexpressed and activated in many cancer types and regulate variable cellular functions in response to extracellular signaling-dependent mechanisms. NRTK-mediated different cellular functions are regulated by kinase-dependent and kinase-independent mechanisms either in the cytoplasm or in the nucleus. Thus, targeting NRTKs is of great interest to improve the treatment strategy of different tumor types. This review deals with the structure and mechanistic role of NRTKs in tumor progression and resistance and their importance as therapeutic targets in tumor therapy.

## 1. Introduction

The expression and activation of protein tyrosine kinases (RTKs) are common in most eukaryotic organisms. These PTKs are involved in the regulation of different cellular processes under normal and pathological conditions [1,2,3]. However, aberrant activation of these PTKs in cancer is mostly associated with disease progression and drug resistance [3,4]. Therefore, PTKs are considered not only as important signaling molecules but also as important drug targets in many diseases, especially cancer [1,2]. The family members of the PTK family are characterized by their diversity in structure and function; in addition, they play an essential role in the regulation of different cellular functions [2,3]. The catalytic kinase domains of PTKs are characterized by the ability to phosphorylate substrate proteins at tyrosine residues. The group of tyrosine kinases is subdivided into subgroups: one of them includes the member of receptor tyrosine kinases (RTKs), while the other subgroup includes the member of non-receptor tyrosine kinases (NRTKs) [4,5]. The main function of RTKs is to rapidly and reversibly phosphorylate protein substrates on their serine, threonine, or tyrosine residues, leading to a significant alteration in protein characteristics, including conformation and interaction to drive different cellular processes [3,4]. The two groups of PTKs include the group of tyrosine kinases and the serine/threonine kinases [6]. Receptor serine/threonine kinases (RSTKs) are extracellular transmembrane proteins with cytoplasmic kinase domains. The extracellular domain serves as a ligand-binding domain, while the cytoplasmic domain serves as a bridge between extracellular signaling receptors for members of transforming growth factor-β to maintain the progression of signal transduction processes [7,8]. Tyrosine kinases are a large family of kinases that include both RTKs and NRTKs that function as “on” or “off” switches in many cellular functions [1,9]. Although both RTKs and NRTKs are similar in function, their structures are different [10]. In contrast to RTKs, the NRTKs lack both transmembrane and extracellular domains [11,12], while RTKs are cell surface receptors that functionally transmit growth factors, cytokines, and hormone signaling to the cytoplasm and nucleus [13,14]. NRTKs are both cytoplasmic and nuclear proteins; the localization of the cytoplasmic NRTK proteins is either free or anchored to the inner side of the cell membrane [15,16]. The main function of the cytoplasmic NRTK protein is to mediate intracellular signals resulting from extracellular receptor-dependent activation [15,17]. 

NRTKs are enzymes with kinase activity whose activation is closely associated with significant changes in annotated molecular and cellular functions [15,16]. Based on their sequence similarity, particularly withing their kinase domains, the NRTKs are grouped into different subfamilies, which are implicated in the regulation of different cellular functions, including cell growth, cell division, and immune regulation [15,18]. In this review, the structure and the mechanistic role of NRTKs in tumor progression and drug resistance are discussed. 

## 2. Structure of Non-Receptor Tyrosine Kinases

In contrast to the RTKs, the members of NRTKs lack both the extracellular ligand-binding domain and the transmembrane-spanning region, which are characteristic of RTKs. The different members of NRTKs are either localized in the cytoplasm or anchored to the cell membrane via significant modification in their amino terminus [15,18].

The most common NRTKs include Ack, Jak, Fes, Fak, Tec, Src, Csk, Abl, and Syk kinases [15,18]. Apart from their classification, NRTKs share distinct sequence similarity within their kinase domains [15,18,19]. These kinase domains are highly conserved, and their catalytic domains are like that of Ser/Thr protein kinases [15,18,19]. 

Of note, Ser/Thr protein kinases are characterized by their lobes on their N- and C-terminal domains. The lobe of the N-terminal domain serves as a binding site for the phosphate groups of adenosine triphosphate (ATP), and the lobe of the C-terminal domain serves as a substrate-binding site for both ATP and peptides [20,21]. The activation loop of the C-terminal domain of NRTKs contains either Tyr, Ser, or Thr residues. The phosphorylation of most NRTKs is mediated by either autophosphorylation or other members of NRTKs at tyrosine sites in the activation loop, and the level of their phosphorylation is mostly associated with significant activity [22,23]. 

Beside the catalytic domains, NRTKs also contain non-catalytic domains. The main function of non-catalytic domains is to regulate the activation of NRTKs via a mechanism mediated by both inter- and intramolecular interactions of PTKs [15,23,24]. The most common non-catalytic domains of NRTKs are the Src homology 2 (SH2) and Src homology 3 (SH3) domains [15,23,24]. The SH2 domain is characterized by its affinity for binding to proline-rich motifs existing in many proteins [15,25,26]. However, the classification of NRTKs into distinct families relies on the molecular analysis of the domain structure based on the variation in amino acid sequences and the genomic organization of the kinase domains [19]. The structure of the most common NRTKs families is outlined in Figure 1.

### 2.1. Ack Family

The Ack is a large protein of 120 kDa whose kinase activity can be mediated by the phosphorylation of its tyrosine residues. The Ack family is comprised of two members, including Ack1, the activated Cdc42-associated kinase 1, and Tnk1, the thirty-eight-negative kinase-1 [27,28,29]. Ack1 is characterized by its unique structure, which differs significantly from that of the other members of NRTKs. In contrast to homologous proteins, the human Ack1/Tnk2 is characterized by its domain architecture that includes an N-terminal sterile alpha motif (SAM) domain [19,30,31,32], an SH3 domain, and a Cdc42/Rac-interactive domain (CRIB), as well as several functionally relevant regions located in the C-terminal region of Ack1 [19,30,33]. These functional regions include multiple proline-rich sequences, a clathrin-binding motif (CB) [34], a region that shares a high homology with Mig6 [19,35], and a ubiquitin binding domain (UBS) [36]. Besides being the only NRTK with the SH3 domain located to the C-terminal of the kinase, the Ack family kinases have the unique CRIB domain [19,30,31]. The catalytic kinase domain of Ack is in the C-terminal region and is closely followed by the SH3 domain, while the SH3 domain is directly followed by the CRIB domain and is characterized by its GTP-dependent binding to Cdc42, but not binding to those of Rho family guanosine triphosphatase (GTPase) [19,30,37]. The homology of both Ack and Tnk ends directly after the SH3 domain (Tnk lacks the CRIB domain that is characteristic of Ack1 [28]). Ack1 contains an arrestin-like clathrin-binding region that is located directly after the CRIB domain [22,38]. However, the multidomain structure of the Ack1 family not only facilitates its localization to different cellular compartments but also promotes its association with different proteins that enhance its functional diversity [22,39]. 

Although the conformation of the activation loop of both the phosphorylated and non-phosphorylated forms of Ack is similar, the similarity of the Ack forms is that they cannot cover the substrate binding site, suggesting that phosphorylation of the activation loop does not appear to play a stimulatory role in the activation of Ack kinase [40,41].

The functional analysis of the Ack family members provided insight into the mechanisms regulating the activation of Ack both in normal and cancer cells [27]. The identification of 10 amino acid sequences rich in proline residues within Ack1 up-stream of the kinase recognition segment of the Mig6 homology region (MHR) has been shown to play an essential in the interaction between the proline-rich sequence and SH3 domain to stabilize the auto-inhibited state of the Ack1 kinase via a mechanism mediated by the MHR-dependent inhibition of the C-terminal region of the kinase domain [22,30,31]. Thus, Ack1 activation is a consequence of the interaction of RTKs with the MHR of Ack1 [22,31,42]. The complete activation of Ack1 has been reported to be attributed to the amino-terminal SAM domain-dependent mechanism, including membrane localization and the symmetric dimerization of Ack1, leading to Ack1 transphosphorylation and the activation of Ack1 [22,27]. To that end, Ack1 is expected to switch to different modes of kinase activation and to be adapted to cellular requirements. 

The identification of the substrates of Ack1 has taken place both in normal and cancer cells. For example, the phosphorylation of Wiskott–Aldrich syndrome protein (WASP) by Ack1 has been reported to promote its actin remodeling activity [28], while the phosphorylation of p130Cas, a component of focal adhesion and a member of the Cas (Crk-associated substrate) family, by Ack1 is associated with tumor invasion, promoting cell spreading [43] and cell migration [44]. In addition to its role in the regulation of cell adhesion and migration, Ack1 has also been reported to phosphorylate the androgen receptor (AR), leading to the enhancement of AR-mediated gene transcription [45,46]. The domain architecture of the human Ack and its homologous proteins in different organisms are shown in Figure 2.

Ack1 resides in the cellular cytoplasm downstream of a broad range of receptors such as integrins, muscarinic M3 receptors, platelet-derived growth factor receptors [PDGFR], Axl receptor tyrosine kinases, insulin receptors (IRs), Mer receptor tyrosine kinases, Trk, and EGFR [34,47,48]. The activation of Ack1 results from the interaction of ligand-activated RTKs with the MHR to overcome the autoinhibitory interaction of the MHR with its own kinase domain [19,22,35]. However, the complete activation of Ack1 is mediated by the SAM domain-dependent mechanism to facilitate membrane localization and the symmetric dimerization of Ack1, leading to its transphosphorylation and activation [22,35]. 

Ack1 kinase as an oncogenic protein relies on the ability of activated Ack1 to promote tumor growth in vivo [49]. The oncogenicity of Ack1 is attributed to the aberrant activation of Ack1 by different mechanisms, including gene amplification, missense mutation of Ack1, and ligand-dependent activation of RTKs [27,50,51]. The gene amplification of Ack1 has been reported in different tumor types, including cervical, ovarian, lung squamous, head and neck squamous cell carcinomas, breast, prostate, and gastric cancers [27,50,51]. Wide genome analysis revealed the existence of both missense and nonsense mutations within the domains of the Ack1 gene [22,52]. 

The functional analysis of Ack1/Tnk2 helped address the mechanisms whereby these kinases regulate tumor growth and resistance to anticancer agents. The main function of activated Ack1 is to transduce the extracellular signals from the stimulated RTKs by their corresponding ligands [27,53]. RTKs are G-protein coupled receptors (GPCRs), such as platelet-derived growth factor receptor (PDGFR), insulin receptor (IR), and epidermal growth factor receptor (EGFR) [53,54,55,56]. In cancer cells, activated Ack1 mediates the phosphorylation of Wwox protein that is thought to play an essential role in triggering the activation of the Akt pathway and androgen receptor (AR), leading to tumor growth [45,46]. Also, the genetic alteration of Ack1 through gene amplification and somatic mutation can cause the neoplastic transformation of a variety of human malignancies [57]. For example, Somatic mutation in the ubiquitin-binding domains (UBDs) of Ack1 has been shown to mediate EGFR regulation in renal carcinoma cells [54,55,56]. The development of tumor resistance by Ack1 is attributed to its function as an epigenetic regulator [27,58,59,60]. Ack1-induced tumor resistance to anticancer agents is mediated through the interaction of Ack1 with the estrogen receptor (ER)/histone demethylase KDM3A (JHDM2a) complex. Consequently, this complex is responsible for the modification of KDM3A via tyrosine phosphorylation, which leads to the regulation of the transcriptional outcome at homeobox A1 (HOXA1) [27,58]. The epigenetic reprogramming of cell cycle genes by Ack1 has been reported to promote breast cancer resistance to CDK4/6 inhibitors [59,61] and the inhibitor of the estrogen receptor modulator, tamoxifen [62,63]. Loss of Ack1 has been shown to be associated with the upregulation of EGFR and to enhance tumor resistance to BRAF inhibitors [64]. 

The inhibition of the Ack1-mediated phosphorylation of the C-terminal Src kinase has been found to counteract prostate cancer resistance to immune checkpoint blockade [60] and to overcome the acquired resistance of EGFR mutant non-small-cell lung cancer (NSCLC) cells to Osimertinib, the inhibitor of EGFR [64]. Importantly, the combination of ACK1 and KIT inhibitors was found to have a positive therapeutic impact on imatinib-resistant gastrointestinal stromal tumors via anti-proliferative and -migration effect-dependent mechanisms [65], as well as being superior to KRAS-mutant-based therapies in NSCLC [66].

In addition to its role in tumor progression and treatment resistance, the involvement of Ack1 in tumor recurrence has been reported in human hepatocellular carcinoma (HCC) [67] and in prostate cancer [68,69].

### 2.2. Jak/Janus Family

The Jak/Janus family includes four members (JAK1, 2, 3 and TYK2) that are quite different from other NRTKs via their two kinase domains in their C-terminal half [11,15,70]. These two kinase domains are regulated so that one is functional, whereas the other serves as pseudo-kinase domain [70,71]. In contrast to the C-terminal region, the N-terminal region of the Jak family contains stretches of highly conserved sequences that are known as JH domains [71,72,73]. These sequences represent the four-point-one, ezrin, radixin, moesin (FERM), and SH2 domains of Jak family members [74,75,76]. The structures of Jaks include Jak1, Jak2, and Jak3 and TYK2, as outlined in Figure 1.

Unlike other receptor tyrosine kinases, the cytokine receptors lack intrinsic protein kinase domains; therefore, the cytokine receptor cannot cause rapid tyrosine phosphorylation of intracellular proteins [77,78,79]. Consequently, the cytokine receptors recruit and activate a variety of NRTKs, including the JAK family, to induce downstream signaling pathways [78,80]. Cytokine receptors comprise two or more receptor subunits, including γ chain and gp130, that are associated with a JAK monomer [81,82]. Upon receptor ligation by a cognate ligand, the receptor subunits are either realigned or oligomerized to recruit and associate JAKs to facilitate their transphosphorylation and subsequently corresponding activation [83,84]. Thus, once the recruiting and activation of JAKs have been completed, they become able to phosphorylate tyrosine residues within the cytoplasmic regions of the cytokine receptors and thereby facilitate the interaction of the downstream adaptor and effector proteins containing SH2 domains with the cytoplasmic region of cytokine receptors [85,86].

The Jak family members Jak1, Jak2, and Tyk2 are omnipresent in mammals, while Jak3 expression is restricted predominantly to hematopoietic cells [87,88]. JAKs are located mainly in the cytosol, and their expression is cytokine receptors-independent [87,88,89]. Because of their close association with cytokine receptors, the members of the Jak family are localized along with their associated receptors either in endosomes or in the plasma membrane [81,82,83,84,85,86,87,88,89].

The link between Jak family members and cytokine signaling is widely reported [89,90,91,92]. The ligation of cytokines, namely interleukin (IL)-2, IL-4, IL-7, IL-9, IL-15, and IL-21, to the corresponding receptors on the surface of target cells induces their auto-activation by transphosphorylation, leading to the activation of Jak1-dependent signaling pathways [93,94,95,96]. The enhancement of Jak3 activation by the abovementioned cytokines results from the binding of Jak3 to the γ chain subunit of the cytokine receptors [95,96,97,98]. Jak1 is activated by many factors, including IL-6, IL-11, oncostatin M, leukemia inhibitory factor (LIF), ciliary neurotrophic factor (CNF), granulocyte colony-stimulating factor (G-CSF), and IFNs. The abovementioned factors-induced stimulation of Jak1 is mediated by the subunit gp130 of cytokine receptors [99,100,101,102], while the activation of Jak2 is mediated by the interactions of Jak with its cognate transmembrane cytokine receptor proteins in response to growth hormone (GH), prolactin (PRL), erythropoietin (EPO), thrombopoietin (TPO), and the family of cytokines that signal through the IL-3 receptor (IL-3, IL-5, and granulocyte–macrophage colony-stimulating factor, as well as the GM-CSF-dependent mechanism) [103,104]. 

Tyk2 is the first member of the Jak family that is involved in the modulation of IFN signaling-dependent mechanisms [97,98]. Although Tyk2 is essential for the modulation of IL-12 signaling-dependent mechanisms, its role in the modulation of IFN-α/β signaling or cytokines, particularly those using gp130, is not essential [104,105,106]. 

In addition to their significant role leading the regulation of immune cells’ growth and fate [107], the JAK-STAT pathway has been shown to play key roles in the regulation of tumor development and plasticity [108]. The structure and conserved phosphorylation sites of the JAK family are outlined as shown in Figure 3. 

### 2.3. Fes Kinases

Feline sarcoma (Fes) and Fes-related (Fer) kinases are a subgroup of NRTKs with similarity to viral oncogenes. These viral oncogenes are derived from feline sarcoma virus (v-FES) and avian Fujinami poultry sarcoma virus (v-fps). Although the ubiquitous expression of Fer is well established, the expression of FES is only limited to myeloid hematopoietic, neuronal, epithelial, and vascular endothelial cells [15,109,110]. 

Fes kinases are characterized by their unique amino-terminal Fes/Fer/Cdc42-interacting protein homology (FCH) domain in addition to constituting three coiled-coil motifs, a central SH2 domain, and a kinase domain that is in the C-terminal region. The FCH domain lies adjacent to a coiled-coil domain that is like the BAR domain that forms a functional unit that is known as the FCH-Bin–Amphiphysin–Rvs (F-BAR) domain [111,112]. 

Although Fes kinase lacks the SH3 domain, the repression of Fes is mediated via the interaction between the SH2 and kinase domain-dependent mechanism [112,113], while the activation of Fes kinase is mediated through the phosphorylation of both Tyr^713^ and Tyr^811^ residues [112,113,114]. The structure and the functional domains of the NRTK Fes family members are outlined in Figure 4.

### 2.4. Fak Family

The Fak family includes Fak that is expressed ubiquitously and other family members (Pyk2, Cak-beta, Cadtk, Raftk, and Fak2), which are expressed mainly in different organs, including the brain, liver, lung, kidney, and hematopoietic cells [114,115]. Fak family members are flanked with extensive N- and C-terminal regions [116,117,118]; the N-terminal region of Fak family members contains the four-point-one, ezrin, radixin, and moesin (FERM) domain that is followed by the kinase domain. FERM is essential for the modulation the interactions of Fak family members with transmembrane receptors [119,120,121]. The FERM domain of Fak can also mediate interactions of Fak with integrins and/or RTKs [122,123]. The C-terminal region of the Fak family possesses a region that is implicated in focal adhesion targeting (FAT) [124,125,126]. The role of paxillin and talin in the modulation of the FAT-dependent targeting of Fak has been reported [127,128,129,130]. 

The involvement of the Fak family in the regulation of different cellular functions and processes has been reported in several studies [121,131,132,133]. Apart from the catalytic activity of FAK in tumor cells, its subcellular localization of Fak in both the cytoplasm and nucleus can be crucial for the transcriptional regulation of chemokines that can promote the tumor microenvironment (TME) to attenuate the antitumor properties of the host [121,132,133]. Accumulated evidence suggests an important role for Fak family members in malignant processes [134]. Several pathological studies indicate an overexpression of Fak protein in many types, including the colon, ovary, pancreas, and prostate [133]. As expected, the overexpression of Fak was found to be correlated with more aggressive and invasive breast carcinomas [133]. Also, PTK2, the gene that encodes Fak, is highly amplified in human breast cancer and is associated with metastasis-free survival [135,136]. The structure and functional domains of Fak family kinases and Src-FAK-dependent signaling pathways are outlined in Figure 5A and Figure 5B, respectively. 

### 2.5. Tec Family

The Tec family is an NRTK that contains five members with high similarity in their functional domains. The members of Tec family kinases are comprised of the Pleckstrin homology domain (PH) and a Tec homology domain (TH) that is divided into the cysteine-rich, Zn2^+^-binding Btk motif and a proline-rich motif [137,138,139]. The PH and TH domains are followed by SH3 and SH2 domains that are consequently followed by the kinase domain [140,141,142]. Unlike other Tech family members, the Bmx and Txk kinases do not constitute the typical structure of most of the Tec family members. Both the TH and SH3 domains of Bmx kinase are known as the TH- and SH3-like domains [143,144]. In Txk, both the PH and TH domains are replaced by a unique cysteine string motif containing serine and threonine sites, which is the common palmitoylation [145,146,147].

Although both Bmx and Tec are widely expressed in different tissues and cell types, the expression of other Tec family members is noted mainly in hematopoietic and non-hematopoietic cells [147,148]. Itk is expressed mainly in T and NK cells, and the expression of Btk is common in B lymphocytes and cells of erythromyeloid lineage, and Txk has been shown to be expressed in T helper Type1 cells and to regulate the production of interferon γ in human T lymphocytes [149,150]. 

The activation of Tec kinases is mediated by membrane targeting via the interaction of their PH domains with phosphatidylinositol (3,4,5)-trisphosphate (PIP3) or with other proteins, as well as in response to the phosphorylation of tyrosine residues within the kinase activation loop [141,151]. The structure and functional domains of Tec kinases are demonstrated in Figure 6.

### 2.6. Src Family

The Src is one of the largest family members of NRTKs that is divided into two classes. One of these classes includes the tyrosine kinases with a broad expression range (e.g., Fyn, Yes), while the other class includes kinases with a limited expression range (e.g., Fgr, Lyn, Hck, Lck, Blk, Yrc) [15]. Src family proteins of vertebrates have a similar structure with a molecular mass ranging from 52 to 62 kDa [152,153]. This group of kinases comprise six distinct functional domains that consist of an N-terminal sequence constitute SH4 region of 15 to 17 amino acids residues, including signals for fatty acid modification sites, followed by a unique domain that possesses a low degree of intra-family homology next to the N-terminal region, followed by the SH3, SH2, and tyrosine kinase domains [152,153,154]. The C-terminal part of Src protein contains tyrosine residues, which are essential for the regulation of Src activity [155,156,157].

The analysis of the protein sequence of the SRC family of kinases (SFKs) revealed the presence of eight members, including BLK, FGR, FYN, HCK, LCK, LYN, SRC, CHK, and YES. The members of SKFs are further divided into two subfamilies: the SRC-related subfamily that is known as Src A, that includes the FGR, FYN, SRC, and YES kinases, while the other subfamily, namely the LYN-related subfamily, known as Src-B, includes the BLK, HCK, LCK, and LYN kinases [158,159,160]. 

The C-terminal of the members of the Src A and its homologous kinase Scr-B is characterized by their function as an endogenous inhibitor for Src family protein tyrosine kinases (SFKs) [158,159,160]. The Csk family of NRTKs includes Csk and Chk (also known as Ctk, Ntk, Chk, Hyl, and Lsk). Although Csk is expressed in all organs, it is predominantly found in the thymus and spleen [15,161], while Chk is expressed primarily in the brain and hematopoietic cells [161,162,163]. In addition to the high homology of the amino acid sequences of the Csk family with those of the Src family, the domain structure of the Csk family includes an SH3 domain, followed by an SH2 domain and a tyrosine kinase domain, evidence of its high similarity with the Src family [161,164]. Apart from their higher structural similarity, the two Scr-A and Scr-B kinases underly several divergences in their structures [15]. For example, the Csk family lacks the N-terminal unique domain the constitutes the fatty acid modification sites. This unique domain is essential to anchor the Src family to the membrane [155,164,165]. In addition to the absence of the C-terminal regulatory site, the Csk family possesses no tyrosine residue in its activation loop [24,161,164]. The structural and functional domains of Src kinases are shown Figure 7.

### 2.7. BRK/Frk Family

FRK is an NRTK Fyn-related kinase and is one of the members of the breast tumor kinase (BRK) family [166,167]. FRK is related to SFKs [166] and known as protein tyrosine kinase 5 (PTK5) that belongs to BRK family kinases (BFKs) [166]. FRK kinases include BRK and SRMS that contain C-terminal regulatory tyrosine and N-terminal myristylation sites [166,167,168]. In addition, the Frk is highly homologous to the Src family in terms of its sequence and structure [169]. Frk presents a highly divergent N-terminal sequence followed by an SH3 domain, an SH2 domain, and a tyrosine kinase domain [168,169]. Unlike the Src family, most of the Frk family members lack the N-myristoylation site [168,170,171], allowing them to be localized to the nucleus but not to the membrane [168]. 

The unique feature of FRK and IYKs is associated with the presence of a nuclear localization signal (NLS) within the SH2 domain [168,169,170,171,172]. This NLS is a bipartite motif that is organized into two clusters of basic amino acids that are separated by a spacer of nine amino acids [171,172]. The two NLS sequences of FRK are organized as follows: the first NLS sequence (KRLDEGGFFLTRRR) is followed by a spacer of nine amino acids extending from amino acid residue 168 to amino acid residue 181, followed by the second NLS sequence (KRLDEGGFFLTRRR) [172,173]. The two NLS sequences of IYK are organized as follows: the first NLS sequence (RRDEGGFFLTRRK) is followed by a spacer of nine amino acids extending from amino acid residue 175 to amino acid residue 188, followed by the second NLS sequence (RRDEGGFFLTRRK) [171,172,173]. 

SFKs are highly homologous in structure, consisting of four consecutive Src homology (SH4, HS3, SH2, and SH1) domains [168,174]. Of note, the SH4 domain serves as a membrane targeting region for myristylation and/or palmitoylation at the N-terminus of SFK members [175]. Thus, the myristylation and/or palmitoylation of SH4 is essential to facilitate the membrane localization of SKF members [176]. The structure of BRK/Frk family members is outlined in Figure 8. 

### 2.8. Abl Family

The two Abl family members include Abl and Arg, which are widely expressed in all organs; while the highest expression level of these is noted in the thymus, spleen, and testes [177,178], the highest expression level of r expression Arg is observed in the brain [176,177]. The structure of Abl is like that of the Src family with an N-terminal region including one SH3, one SH2, and one tyrosine kinase domain [179,180]. The Abl family kinase lacks a C-terminal-negative regulatory site that is characteristic of Src family members [181,182] and instead possesses a C-terminal region that contains an F-actin-binding domain (FABD) [183,184,185,186]. The kinase domain is followed by a proline-rich DNA-binding domain [184,185,186]. The C-terminal region of Abl represents an actin-binding domain containing three nuclear localization signals (NLSs) and one nuclear export signal (NES) [187].

Arg shows high sequence similarity to Abl within the N-terminal region [188,189,190,191]. The sequence similarity of Arg within the C-terminal region is moderate compared to that of Abl [190,191]. However, the critical elements of Arg structure include the conserved SH3-binding sites and the actin-binding domain [192,193]. 

The mechanisms regulating the subcellular localization of Abl to the membrane are mediated by its own structural domains [194,195,196,197]. The common subcellular localization of Abl in different cell types localizes to the nucleus and cytoplasm, where they are mostly membranes and actin filaments [186,194,195,196,197]. The localization of Abl in hematopoietic cells and neurons is common in the cytoplasm, while the localization of Arg appears to only be cytoplasm [195,196,198]. The nuclear localization of Abl is determined by both NLS and NES sites [197], while its membrane localization is mediated via a myristoylation site-dependent mechanism [182,198], and the binding of Abl to the actin cytoskeleton is mediated via an actin-binding domain-dependent mechanism [182,199]. 

The activation of ABL family members (e.g., Abl1, ARG) is mediated by different mechanisms, including mutation, dimerization, and phosphorylation [200,201,202], and by the phosphorylation of the tyrosine 412 (Y412) and tyrosine 245 (Y245) residues within kinase and interlinked regions, respectively [203,204,205,206]. However, the constitutive activation of Abl by the disruption of its autoinhibition mediated by the translocation to a variety of many genes such as BCR, Tel, and ETV6 is the main cause of the development of different tumor types, including leukemia [207]. Although the role of both c-Abl and Arg in the development of a variety of human leukemias has been discussed, their oncogenic function in solid tumors has not been determined [208]. However, data from the immune histochemistry analysis of c-Abl and/or Arg show that both Abl and Arg are overexpressed in some solid tumors, including the brain, lung, ovarian, colon, and prostate cancer. In addition, c-Abl amplification was found in renal medulla carcinomas [209,210,211]. The structure and functional domains, as well as the 3D of ABL kinase protein, are outlined in Figure 9A,B.

### 2.9. Syk Kinase

The spleen tyrosine kinase (Syk) is localized mainly in different tissue types [15,212,213,214,215,216]. The members of Syk family kinases include Syk and Zap70, which are similar in terms of their structure, and they constitute two SH2 in tandem followed by a catalytic domain in the N-terminal half and catalytic kinase domain in the c-terminal region [217,218,219]. The members of the Syk family do not possess any fatty acid modification sites, and consequently, they are recognized as cytosolic proteins [220]. The localization of the members of the Syk family is dramatically affected by cell stimulation [221,222,223]. The SH2 tandem domains of the Syk family have been reported to exhibit extremely high affinity for doubly phosphorylated tyrosine-based immune receptor activation motifs (ITAMs) [217,223]. The aim of the phosphorylation of ITAMs by Src family PTK is to initiate signaling events for multiple receptors and subsequently generate an excellent binding platform for the Syk family [224,225]. Consequently, the cytosolic molecules of Zap and Syk can be translocated to the cytoplasmic tails of immune recognition receptors (MIRRs) in the proximity of their membrane [225,226]. The structural and functional domains of the Syk kinases are demonstrated (Figure 10).

## 3. Mechanisms of NRTK-Mediated Pathways in Normal and Cancer Cells 

Unlike RTKs, which are transmembrane receptors, NRTKs are in the cytoplasm and play an important role in the regulation of various cellular functions through the modulation of intracellular signaling [11,12]. The NRTK family members are distributed among the different subcellular compartments [11,12] and are determined by either factor on the cell surface or by integrating heterologous protein–protein interactions in the cytoplasm [15,227]. 

As an intracellular cytoplasmic protein that transduces intracellular signaling, NRTKs have the potential to play an essential role in the modulation of signaling pathways such as the JAK pathway, which has the potential to regulate gene expression via the IL-6-mediated phosphorylation of membrane-bound tyrosine kinase (TK) and Janus kinase that is responsible for the activation of the signal transducer and activator of transcription (STAT) [15,228]. In addition to its role in the inhibition of cell growth and induction of apoptosis [13,18], Abl1 plays a central role in the regulation of tumor development and progression [229]. In human leukemias, the enhancement of the oncogenic activity of Abl is mediated by chromosomal translocation [t(9;22) (q34;q11)] to generate BCR-ABL1 fusion proteins, leading to the disruption of intramolecular interactions of inhibitory Abl1 [230]. The oncogenic activation of Abl1 in BCR-ABL1 chimeric proteins is mediated by an N-terminal coiled-coil (CC) oligomerization-dependent mechanism [226]. Also, the involvement of multiple signaling pathways such as RAS, NF-κB, PI3K-Akt, Jun, β-catenin, and STAT is considered [231].

The regulation of cellular functions in both normal and tumor cells by the members of NRTKs is common. For example, Fak has been reported to trigger the regulation of cell adhesion and proliferation [232,233], besides being one of the most important components of the signal transduction pathways of Fyn [234]. Also, the involvement of the Ack family kinases in the regulation of cell growth is associated with the induction of both JAK and Src-dependent mechanisms [235,236]. The Tec family of protein tyrosine kinases plays an important role in cellular signaling through an antigen-receptor-dependent mechanism such as the TCR, BCR, and FCε receptor [237,238]. Syk functions as a mediator for the immune response between cell receptors and intracellular signaling [218,220].

In contrast to the activation of RTKs that is determined by ligand-binding-dependent mechanisms, the activation of NRTKs is much more complex and is mediated by a heterologous protein–protein interaction, a mechanism that is essential to initiating signal transduction processes via a transphosphorylation-dependent mechanism [1,15,175]. 

The oncogenic activity of Abl1 in the context of the ABL1-BCR fusion protein is mediated via the BCR N-terminal coiled-coil (CC) oligomerization domain [208,239,240]. The activation of both c-Abl and Arg kinases in human primary melanomas is associated with melanoma cell invasion via a mechanism mediated by Stat3 [209,241]. Moreover, the co-activation of both c-Abl and Arg kinases in solid tumors was found to enhance platelet-derived growth factor (PDGF)-induced epithelial–mesenchymal transition and promote transforming growth factor-β (TGF-β)-induced tumor growth [15,211]. Also, the ability of Abl1 kinase to phosphorylate AKT and ERK and to promote cancer cell proliferation and survival has been reported [242].

The role of Abl1 in the development of HCC is mediated by the c-MYC/NOTCH1 axis [239]. Thus, the inhibition of Abl1 has been shown to be a promising therapeutic strategy to treat HCC in patients with overexpressed Abl1. Abl1 kinase inhibitors have been approved for their clinical relevance in many tumor types, including leukemia and HCC, in patients showing the overexpression and/or activation of Abl1 [243,244,245].

The JAK signal transducer and activator of the transcription (JAK-STAT) pathway are an evolutionarily conserved mechanism of transmembrane signal transduction that enables cells to communicate with their exterior environment [70,246]. The activation of the JAK-STAT pathway is regulated by various factors, including cytokines, interferons, and growth factors [70,247]. Consequently, the JAK-STAT pathway has the potential to drive physiological and pathological processes, leading to the regulation of cell proliferation and metabolism in addition to inducing immune responses and enhancing inflammation and malignancy [70,247]. Thus, the dysregulation of JAK-STAT signaling in combination with related genetic mutations is strongly associated with immune activation and tumor progression [248,249]. 

Fes kinase is characterized by its oncogenic properties that have been confirmed both in cell lines and in animal models [250,251]. FES kinase has a limited expression pattern in both endothelial and epithelial cells, particularly those of neuronal origin, and in cells with hematopoietic origin, such as myeloid cells [15,250,251]. The therapeutic impact of Fes in tumor therapy relies on evidence suggesting the role Fes kinase in the regulation of tumor microenvironment-dependent mechanisms and its functional role as a tumor suppressor protein in melanoma cell lines [15,252]. Moreover, targeting Fes-related kinase (Fer) was found to have an inhibitory effect on melanoma growth and metastasis [253,254]. Beyond its impact as a therapeutic target in melanoma, the overexpression of Fer is correlated with cancer cell survival in non-small-cell lung cancer (NSCLC) as well as in lung adenocarcinoma cell invasion and tumor metastasis, besides being a prognostic marker for poor prognosis [255,256]. The loss of Fer has been shown to impair metabolic plasticity both in lung and breast carcinoma cells [257]. Also, the role of Fer kinase in the regulation of the mechanisms of HCC metastatic dissemination has been reported [258]. 

The involvement of integrins in the regulation of many fundamental cellular processes, including cell adhesion and migration, has been widely reported [248,249]. As transmembrane adhesion receptors, integrins are localized at cell–matrix contact sites, where they connect components of the extracellular matrix (ECM) and interact with several structural and signaling molecules, such as talin, paxillin, vinculin, ά-actin, FAK, and Src [248,249]. Thus, as integrin mediators, both FAK and Src kinases have the potential to have cell adhesion-dependent responses by direct and indirect interaction with integrins [250,251]. The direct interaction of ß3 integrins via their cytoplasmic tail with the SH3 domain of Src kinase results in the activation of Src that in turn triggers the activation of the guanine nucleotide exchange factor (GEF) and T cell lymphoma invasion and metastasis 1 (Tiam1) [250,251,252]. These factors together have been reported to be essential for the activation of Rac that can then induce the stimulation of actin-driven positive activity at the site of integrin binding, finally leading to cell adhesion or migration [250,251,252].

Also, the interaction of ß3 integrin with the SH3 domain has been shown to trigger the stimulation of STAT3 and FAK signaling, leading to tumor growth [253,254]. The phosphorylation of ß1 integrin is required to inhibit Rho-mediated cytoskeletal contractility in addition to mediating the regulation phenotype transformation [253,254]. Aside from its contribution in the modulation of several cellular functions, Src can also weaken the connections between ECM, integrins, and cytoskeleton, leading to cell adhesion turnover and remodeling of the actin cytoskeleton, suggesting an important role for FAK in the modulation of integrin signaling [253,254,255,256]. For example, FAK has been reported to act as a signaling molecule and scaffold and recruit Src and Src substrates to integrin sites [255,256,257,258]. Consequently, the cell–matrix contact-dependent clustering of integrins is expected to be the main mechanism leading to the activation of FAK, which has been reported to result from the cell–matrix contact-dependent clustering of integrins [259,260]. 

## 4. NRTKs as Therapeutic Target in Tumor Treatment

NRTKs are key players in intracellular signaling pathways, and their dysregulation in the form of constitutive activation is expected to be associated with tumor initiation, progression, and drug resistance. Thus, targeting aberrantly activated NRTKs may prevent tumorigenesis in addition to having an impact as a therapeutic target for anticancer agents. 

The analysis of BRAF mutant melanomas revealed a contributing role for Abl kinase activity in the enhancement of acquired resistance to BRAF and MEK inhibitors in therapy-resistant cells [261]. Abl kinases are expected to play an important role in the development of drug resistance in cancer since the suppression of Abl1/Abl2 activity by selective kinase inhibitors Nilotinib or genetic inhibitors was found to sensitize cancer cells to BRAF and MEK inhibitors [257]. Some clinically available tyrosine kinase inhibitors, particularly those targeting Abl kinases, have been approved in BCR-Abl-driven chronic myelogenous leukemia (CML) [262], as well as in many solid tumors, including lung cancer [263,264]. 

As such, Fes has been suggested to exert both tumor-stimulatory and tumor-suppressive effects on cancer cells based on the type of malignancy [265,266,267,268]. 

Although Fes represents a potential therapeutic target in different tumor types, its inhibition in macrophage and vascular endothelial cells attenuates their tumor-promoting roles [269,270,271,272]. 

JAK/STAT is one of the versatile signaling pathways whose crucial role in tumor progression has been extensively studied. The potential interaction of JAK/STAT with many signaling pathways suggested it as a potential therapeutic target for the development of clinically relevant small molecular inhibitors. The abnormal activation of JAK/STAT signaling is frequently common in a variety of different tumor types. 

Activation of the JAK-STAT pathway is mediated by the ligation of growth factor receptors with the corresponding ligands, leading to receptor dimerization and the recruitment of related JAKs. Activated JAK then becomes able to phosphorylate tyrosine residues of the receptors to form docking sites for the phosphorylation of STATs and other substrates to trigger abnormal cell proliferation tumor initiation, progression, and drug resistance [273,274,275,276]. In skin cancer, particularly melanoma, the activation of the JAK/Stat pathway is mostly associated with tumor progression and treatment resistance [277,278]. Melanoma has been reported to display distinct gene expression profiles that depend on the activation levels of individual STAT proteins [279]. Other studies indicate that the phosphorylation of STATs may function as a predictive marker for melanoma progression [278,280]. The alteration of STAT proteins both in melanoma cells and immune cells of melanoma patients has been reported [281,282]. Thus, targeting the JAK/STAT pathway is expected to have a therapeutic impact on melanoma treatment [283,284]. 

In addition to their involvement in various signaling networks, the JAK/STAT pathway is continuously in crosstalk with different pathways, particularly those associated with hematological malignancies and melanoma [285,286]. 

The evaluation of the JAK/STAT pathway as a marker for esophageal squamous cell carcinoma (ESCC) progression has been confirmed in tumors and adjacent normal esophageal epithelia by immunohistochemistry [286]. Also, the increased activation of p-JAK1 and p-STAT3 in primary tumors revealed their reliability as a predictive marker for poor prognosis in ESCC patients [287]. Thus, beside serving as a predictive marker with a poor prognosis and playing a crucial role in the regulation of ESCC progression, the JAK/STAT pathway is a promising therapeutic target for tumor treatment [288,289]. Targeting JAK2 by its specific inhibitor, AG490, was found to decrease the inflammation and development of ESCC via the mechanism mediated by the suppression of STAT3 [290,291]. 

In HCC, the phosphorylation of STAT3 has been reported to target growth factors and cytokines like TNF, IL-6, hepatocytes growth factor (HGF), and epidermal growth factor (EGF) family [292,293]. However, the aberrant activation of SAT3 in response to the mentioned stimulators has been shown to be a JAK1-depnednt mechanism and is associated with HCC progression and metastasis [294,295]. 

The tumorigenic role of JAK/STAT signaling has been reported in different tumors including head and neck squamous cell carcinoma (HNSCC) [296,297] and in breast cancer [298]. In addition, JAKs and STATs were found to contribute to the development of breast cancer by acting as oncogenes [299,300]. Analysis of breast cancer specimens revealed that JAK1, JAK2, and JAK3 are frequently targeted for somatic mutations, which may be the main cause of the development of breast cancer [301,302,303]. 

Ack1 has been shown to have a therapeutic impact on various tumor types, including breast cancer [304], HCC [305,306,307], and colorectal cancer [308,309,310]. The inhibition of Ack1 alone was found to inhibit proliferation and migration in many tumor types [305,306,307] and to enhance the apoptosis of lung cancer cells [311].

Syk is a modulator of tumorigenesis that can drive opposite functions. In addition to having tumorigenic activity in some cells, Syk can also act as a tumor suppressor in other tumor cells [312]. In hematopoietic malignancies, Syk is mostly associated with tumor progression and survival, while its inhibition or silencing is frequently associated with the induction of apoptosis [313,314,315,316,317]. In cancers of non-immune cells, in addition to initiating a pro-survival signal, Syk suppresses tumorigenesis via mechanisms mediated by the restriction of epithelial–mesenchymal transition, improvement in cell–cell interactions, and inhibition of migration [318,319].

The activation of Syk is mediated by the binding of its SH2 domains to the immunoreceptor tyrosine-based activation motif (ITAM) that is known as a common sequence of amino acids located on most SYK-coupled receptors [320,321].

Thus, targeting the Syk signaling pathway is a promising therapeutic target for diseases such as cancer. Entospletinib, for example, is one Syk inhibitor that has shown promising therapeutic effects against B cell malignancies in clinical trials [322]. In addition, some combinational regimens based on Syk inhibitors have been evaluated in several malignancies [323,324,325]. 

IL-2-inducible kinase (ITK) is a member of the Tec kinase family that is essential in driving T cell receptor (TCR) signaling [326]. ITK kinase has been reported to play an essential role in the development of acute lymphoblastic T cell leukemia and cutaneous T cell lymphoma [327]. Thus, in addition to being a promising therapeutic target for other human diseases, ITK kinase has been approved as an effective therapeutic target for T cell malignant lymphoma [327].

Targeting Fak was found to enhance the therapeutic efficacy of both radio- and chemotherapy-based treatments [324,328]. Most chemotherapy drugs, particular platinum-based agents and fluoropyrimidine chemotherapy as well as radiation, mediate their anticancer effects in cancer cells via DNA damage-dependent mechanisms [329,330]. 

In addition to its crucial role in the regulation of DNA damage and repair mechanisms, Fak can also coordinate the nuclear translocation of β-catenin to enhance the transcription of genes involved in the regulation of DNA repair mechanisms to promote cell survival and drug resistance [331,332]. Therefore, targeting FAK is a promising strategy to restore sensitivity to DNA damage therapy [324,333]. 

The role of Src kinase in the regulation of tumor growth and metastasis has been reported [334,335]. The involvement of Src in the regulation of tumor cell-induced bone destruction has been shown to be an osteoclasts-dependent mechanism since Src has been reported to enhance osteoclast activation and bone resorption, mechanisms that are thought to be associated with the progression of bone metastases [336].

Src inhibitors have been approved for their ant tumor activities in different tumor types, including osteoporosis and osteolytic bone disease [337]. The most common Src inhibitors include NVP-AAK980, an ATP target compound [236], AZD0530 (Saracatinib) [338], AP23451 [339], and the multi-targeted kinase inhibitor Dasatinib (BMS-354825). These can all effectively inhibit ABL-BCR, Src family, and other kinases via a mechanism mediated through the destruction of the ATP-binding site of the enzymes [340]. However, the most used clinical inhibitors are Saracatinib and Dasatinib inhibitors [341].

Csk is the member of the SFK family and has the potential to mediate its inhibitory effects on SFK activity by the phosphorylation of its C-terminal Tyr^527^ residue of the negative regulatory loop [342]. Csk has the potential to trigger tumor suppression through the inhibition of the oncogenic activity of SFKs [342]. The overexpression of Csk has been shown to inhibit the growth of human colon cancer cells [343]. In addition to its inhibitory effect on tumor growth, the loss of Csk was found to be associated with the development of HCC in a murine model [344]. 

Several kinase inhibitors targeting family members have been further developed and approved for their clinical application. The most common clinically applied inhibitors include imatinib/(STI-571), Dasatinib/(BMS-354825, Nilotinib/(AMN107), Bosutinib (SKI-606), Radotinib6/(IY-5511), Ponatinib/(AP24534), Asciminib/(ABL001), Ibrutinib/(PCI-32765), Acalabrutinib/(ACP-196), Zanubrutinib/(BGB-3111), Ruxolitinib/(INC424), and Fedratinib/(SAR302503, TG101348). 

Imatinib is an Abl tyrosine kinase inhibitor (TKI) with high affinity to bind to the ATP-binding pocket of the Abl kinase domain and has the potential to BCR-ABL kinase [341]. Therefore, a mutation in the threonine 315 residue (T315) of the ATP-binding pocket is critical for the accessibility of the ATP-binding pocket and can confer resistance to imatinib and the related TKIs [341]. Second-generation BCR-ABL inhibitors such as Nilotinib/(AMN107) have a clinical utilization which has confirmed the therapeutic potential of second-generation TKIs to overcome imatinib resistance caused by ABL kinase point mutations [341]. Of note, Nilotinib is an ATP-competitive type II kinase inhibitor with a greatly improved potency compared to imatinib [341,342]. 

Bosutinib belongs to the second-generation of TKIs besides being characterized with its dual inhibitory activity against SRC and ABL kinases [341,342,343,344]. Bosutinib inhibitor has minimal inhibitory effects against other tyrosine kinases such as c-KIT and PDGFR-β kinases, but it is more effective against imatinib-resistant leukemias [341,342,343,344]. The inhibitory effect of Bosutinib was more pronounced in patients with Philadelphia chromosome-positive chronic myeloid leukemia (Ph+ CML), particularly in patients who are resistant to treatment with imatinib [345]. Other agents such as Radotinib, that is clinically approved as a second-generation BCR-ABL inhibitor, can exhibit inhibitory effects on some imatinib-resistant tumors [346]. Also, the allosteric inhibitor Asciminib has been reported for its affinity to bind to the myristate pocket of BCRABL and is thereby effective against T315I-mutant BCR-ABL [347]. The third-generation inhibitor, Ponatinib, is clinically approved for the treatment of both wild-type and T315I-mutant BCR-ABL [341,348]. 

Ponatinib is a third generation of TKI that is characterized by its inhibitory effects on multiple kinases besides being an effective BCR-ABL1 inhibitor with higher activity against the ABL kinase domain with mutations such as tyrosine315 isoleucine mutation (T315I) [348,349]. The antitumor activity of Ponatinib as TKI relies on its ability to inhibit the catalytic activity of native and mutant ABL [350,351]. In addition to the unique, multi-targeted properties of Ponatinib, a number of studies have demonstrated its ability to target other important tyrosine kinases (FGFR, PDGFR, SRC, RET, KIT, and FLT1) in different human malignancies [350,351].

Thus, based on its functional role in tumor progression SFK is a potential therapeutic target [351]. The potential of Dasatinib to target multiple kinases, including Src, Fgr, Fyn, Hck, Lck, Lyn, and Ye, has been widely documented [341,350,351]. 

## 5. Non-Receptor Tyrosine Kinases and Protein Tyrosine Phosphatases

The regulation of the activity of NRTKs is mediated by protein tyrosine phosphatases (PTPs), leading to the dephosphorylation of protein targets and adopting the activity of protein kinases in response to physiological condition [11,352]. PTPs play an important role in the regulation of tyrosine phosphorylation and signal transduction [353,354].

These PTPs consist of functionally conserved PTPase domains with highly conserved signature motifs (His/Val) Cys(X)5Arg (Ser/Thr), which are characterized by their catalytic function, leading to hydrolyzing phosphate from the tyrosine residues of their substrates [5]. PTPs are tyrosine-specific phosphatases or dual-specific phosphatases [355,356]. The main function of specific PTPs is to dephosphorylate only tyrosine residues [355,356], while the function of dual-specific PTPs is to dephosphorylate both tyrosine and serine/threonine residues [355,356]. However, cellular localization PTPs can also be classified into receptor or non-receptor PTPs receptor types referred to PTPs that span the cell membrane, while non-receptor PTPs are referred to PTPs that are localized mainly in the cytoplasm [357,358]. In addition to their abovementioned classifications, PTPs can be classified into tumor-suppressive or oncogenic PTPs based on the type of dephosphorylated protein kinase of interest [359,360]. Src homology-2-containing protein tyrosine phosphatase 2 (SHP-2) is a non-receptor, tyrosine-specific PTP that binds to the intracellular domain of RTKs via phosphotyrosine residues [361,362,363]. SHP-2 can dephosphorylate proteins or residues that are able to suppress the function of Ras and SFK inhibitors to enhance the activity of tumorigenic pathways such as PI3K and MAPK/ERK signaling [363,364].

## 6. Conclusions

Based on their profound contribution to the regulation of numerous vital cellular mechanisms, non-receptor tyrosine kinases (NRTKs) play an essential role in tumor development, progression, and treatment resistance. Thus, targeting NRTKs is a relevant strategy that can prevent tumorigenesis and its consequences. Also, the clinical use of tyrosine kinase inhibitors, in combination with conventional treatments, allows for targeted-based cancer therapies using small specific inhibitors, which are less toxic than traditional cytotoxic chemotherapy. Therefore, the establishment of effective therapeutic strategies based on the dysregulation of NRTK molecular functioning is essential for patient care.

## Figures and Tables

**Figure 1 cancers-16-02754-f001:**
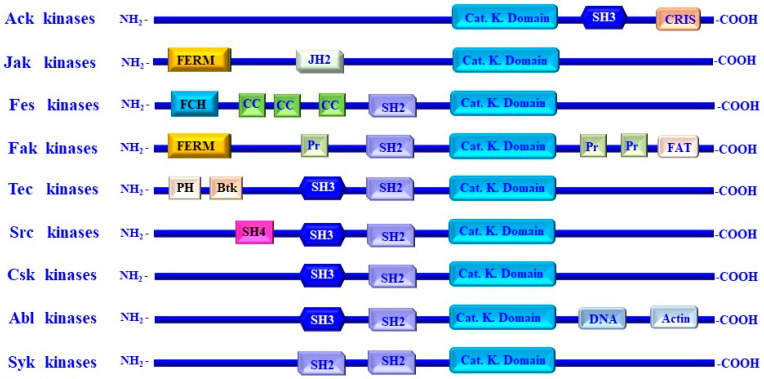
Structure of non-receptor tyrosine kinase families and their functional domains. Non-receptor tyrosine kinases (NRTKs) including Ack, Jak, Fes, Fak, Tec, Src, Csk, Abl, and Syk kinases. NRTKs consist of a single protein with N- and C-terminal regions. The N-terminus contains a kinase domain, which extends over approximately 300 residues, while the N-terminus of NRTKs is larger than the N-terminus. The structure and the functional domains and regions of different members of NRTKs include the following: Src homology (SH) domains that are referred to as SH4, SH3, SH2, and catalytic SH1 domains, the Pleckstrin homology (PH), four-point-one, ezrin, radixin, moesin (FERM), the Janus homology 2 (JH2) domain, and the Fes/Fer/Cdc-42-interacting protein homology (FCH) domains, Bruton’s tyrosine kinase (Btk)-like zinc finger, the coiled-coil motifs (CC), and proline-rich region (pr). The Cdc42/Rac-interactive (CRIB) domains, DNA-binding domains (DNA), action-binding domain, and a focal adhesion targeting domain (FAT).

**Figure 2 cancers-16-02754-f002:**
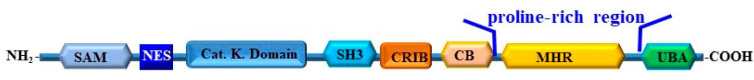
An overview of the domain organization of Ack1. The different structural domains of Ack1 are shown. These domains include the sterile alpha motif (SAM); the nuclear transport signal (NES); catalytic kinase domain (Cat. K. domain); Src homology domain 3 (SH3); Cdc42; Rac-interactive binding domain (CRIB); Cdc42 binding domain (CB); CL, clathrin-binding domain (CB), proline-rich region, Mig6 homology region (MHR); UBA, ubiquitin association domain (UBA).

**Figure 3 cancers-16-02754-f003:**
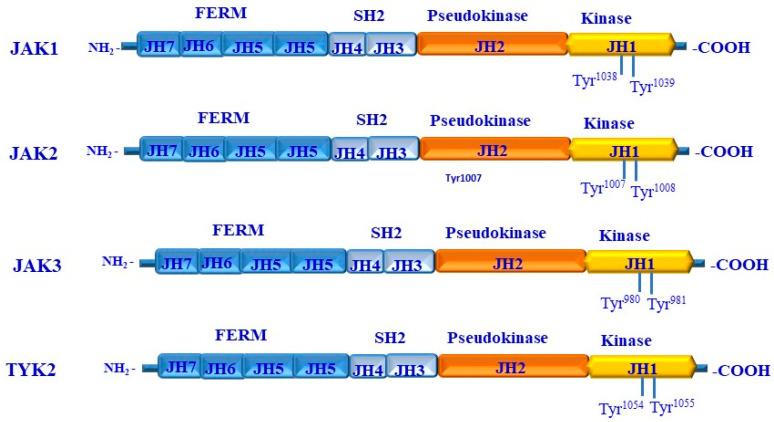
Structure and functional domains of JAKs. Each member of the JAKs (JAK1, JAK2, JAK3, and TYK2) consists of seven homology domains (JH) organized into four different domains, including the kinase domain that constitutes the JH1, the pseudo domain that constitutes JH2, the Src homology2 (SH2) that constitutes both JH3 and JH4, and the four-point-one, ezrin, radixin, and moesin (FERM) domain that constitutes JH5, JH5, JH6, and JH7. The phosphorylation of JAK1 occurs at Tyr^1038^/Tyr^1039^ residues; the phosphorylation of JAK2 occurs at -Tyr^1007^/Tyr^1008^; the phosphorylation of JAK3 occurs at Tyr^980^/Tyr^981^ residues; and the phosphorylation of Tyk2 occurs at Tyr1054/Tyr1055 residues.

**Figure 4 cancers-16-02754-f004:**
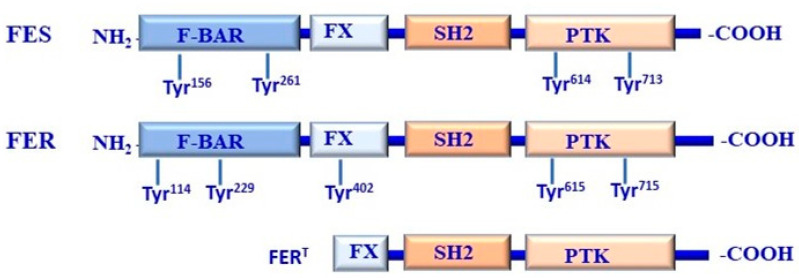
Structure and functional domains of FES/FER kinases. FES and FER share a similar domain organization, including F-Bin–Amphiphysin–Rvs (F-BAR), F-BAR extension (FX), Src homology2 (SH2), and protein tyrosine kinase (PTK) domains. Fes is structured as a dimer mediated by its F-BAR domain that can exist in higher-order oligomers. The N-terminal domain of Fes is to repress its kinase activity, which can be attenuated by proline insertion mutations, leading to the destruction of oligomerization. As a consequence, the repressed Fes oligomer undergoes significant conformation via the interaction of the F-BAR domain with the phosphatidylinositol 4,5-bisphosphate or FX domain binding to phosphatidic acid, which can be produced by phospholipase *D.* Accordingly, Fes more readily interacts with SH2 ligands, which leads to kinase activation via the SH2–PTK interface.

**Figure 5 cancers-16-02754-f005:**
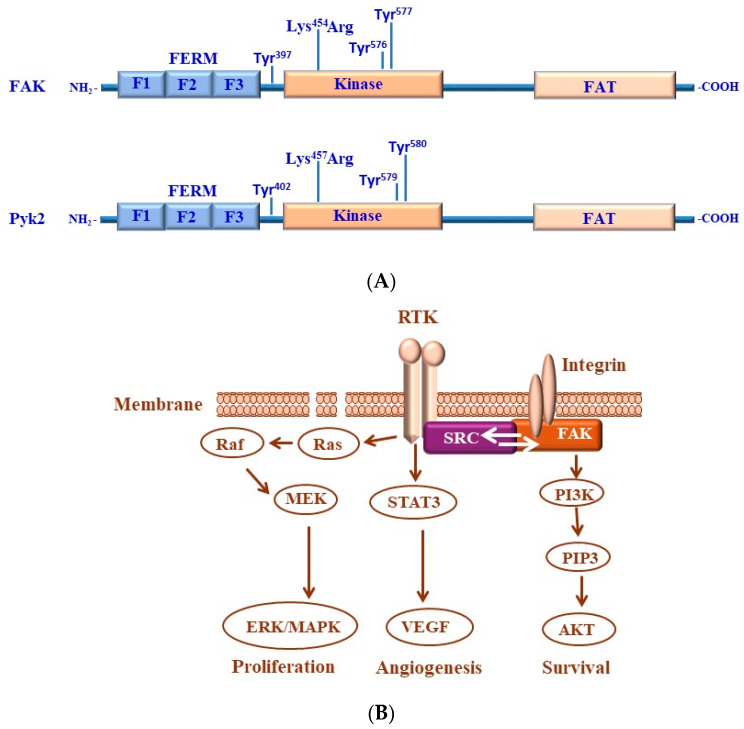
(**A**) Structure of focal adhesion kinase (FAK) and its homologous FAK-related proline-rich tyrosine kinase 2 (Pyk2). Both FAK and PyK2 kinases consists of three domains, including the four-point-one, ezrin, radixin, and moesin (FERM) that is divided into three subdomains, referred to as F1, F2, and F3, central kinase domain, and focal adhesion targeting (FAT) domain. Both FAK and Pyk2 kinases contain a nuclear localization sequence (NLS) and a nuclear export sequence (NES) in addition to several shared conserved phosphorylation sites of tyrosine (Tyr) residues that are in the FAK protein sequence at Tyr^397^, Tyr^576^, and Tyr^577^ residues, while in PyK2, the phosphorylation sites of tyrosine are located at Tyr^402^, Tyr^579^, and Tyr^580^ residues. Both FAK and PyK2 kinases possess a lysine (Lys) mutation in the kinase domain of FAK at Lys^454^ residue and in the kinase domain of Pyk2 at Lys^457^ residue. (**B**) Src/FAK-mediated transduction pathways contribute to cancer progression. FAK is activated by integrins to enhance the activity of phosphoinositide 3-kinase (PI3K). The activation of FAK-dependent pathways is associated with the stimulation of cell proliferation, angiogenesis, and survival. Upon the phosphorylation of FAK, we are able to recruit Grb2 and the p85 regulatory subunit of PI3K to enhance the activation of both the Grb2/Ras/MAPK and PI3K/Akt pathways, respectively, to promote tumor angiogenesis, survival, and proliferation.

**Figure 6 cancers-16-02754-f006:**
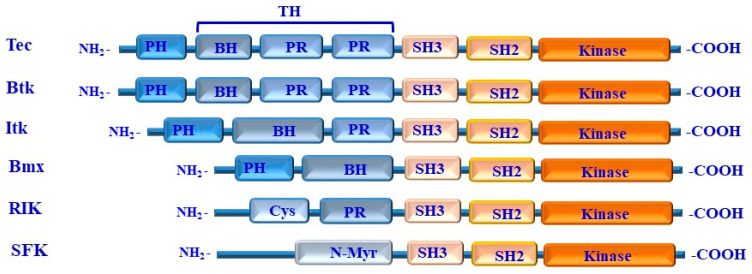
Structure of TEC family kinases. The five kinases of the TEC family are represented by a diagram showing their structural domains. PH: Pleckstrin homology domain; TH: Tec homology domain; BH: Btk homology motif PR: proline-rich region; Cys: cysteine-rich sequence; N-Myr: N-terminal myristylation signal; SH: Src homology domain (SH3, SH2, and kinase) domains.

**Figure 7 cancers-16-02754-f007:**
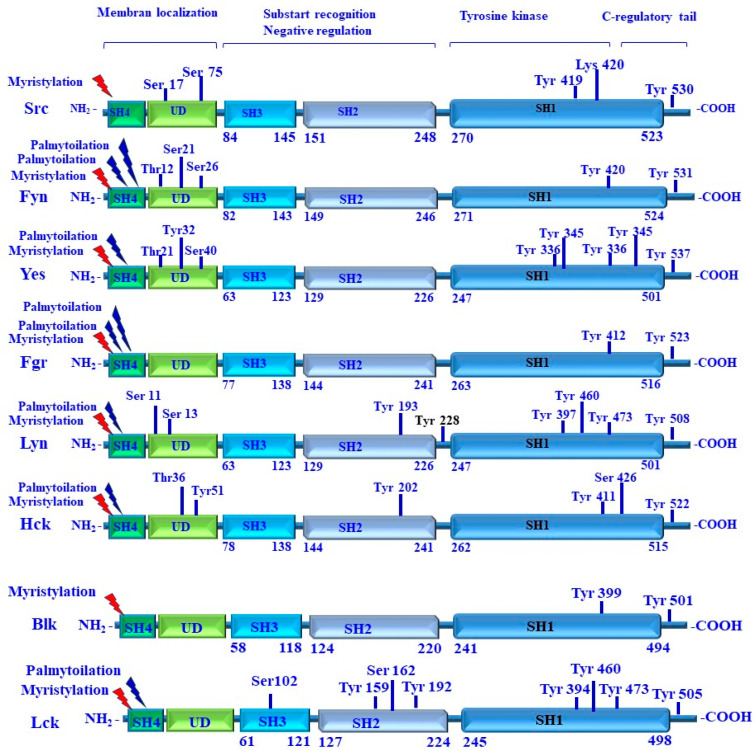
Structure of SRC family kinases, including their functional domains and dependent signals. Activation of Src family kinases is a membrane-dependent mechanism and is regulated by lipid/myristate modification within the SH4 domain and membrane binding. The modification of SH4 by myristylation and/or palmytoilation is to facilitate the localization of Src family kinases to the cell membrane, and the phosphorylation of tyrosine residues within the kinase domain to enhance Src activity. The inhibition of Src family kinases is mediated via ubiquitination sites, lysine 429 (Lys^429^) residue, and/or by the phosphorylation of tyrosine (Tyr) residues located in C-regulatory tail. Domain/regulatory regions are shown as boxes: Src homology 1 (SH1); tyrosine kinase/catalytic domain; Src homology 2 (SH2); Src homology 3 (SH3); Src homology 4 (SH4); and unstructured unique domain (UD).

**Figure 8 cancers-16-02754-f008:**
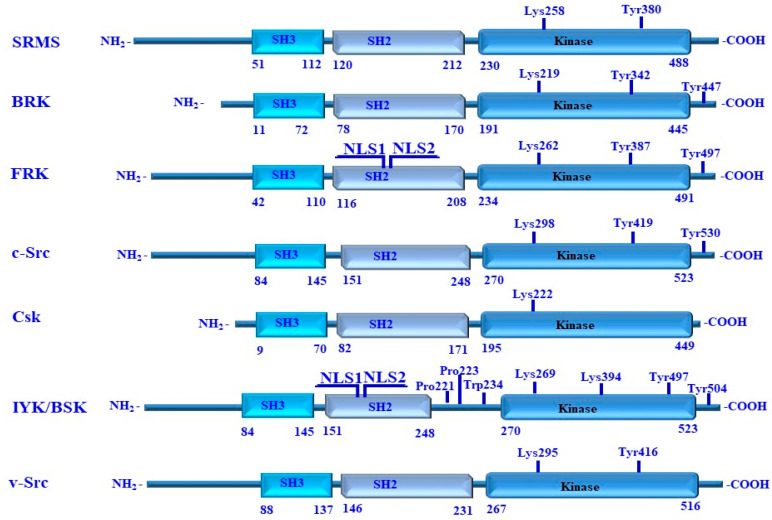
Structure and domains of human SRMS, BRK, FRK, c-Src, Csk, IYK/BSK, and chicken v-Src. These kinases constitute SH3, SH2, and the kinase domain on their N-terminal region. Key residues, which are essential for the regulation of the enzymatic activity of these kinases, are highlighted. These include SRMS Tyr^380^ of the SRMS, Tyr^342^ of BRK, Tyr^387^ of FRK, Tyr^419^ of c-Src, Tyr^497^ of IYK/BSK, and Tyr^416^ of v-Src. The C-terminal regulatory tyrosine residues include Tyr^447^ of BRK, Tyr^497^ of FRK, Tyr^530^ of c-Src, and Tyr^504^ of IYK/BSK. The ATP-contacting lysine residues include Lys^258^ of SRMS, Lys^219^ of BRK, Lys^262^ of FRK, Lys^298^ of c-Src, Lys^222^ of Csk, Lys^269^ of IYK/BSK, and Lys^295^ of v-Src Lys^295^. All numbering is for human proteins, and v-Src is from a retroviral source with its own numbering.

**Figure 9 cancers-16-02754-f009:**
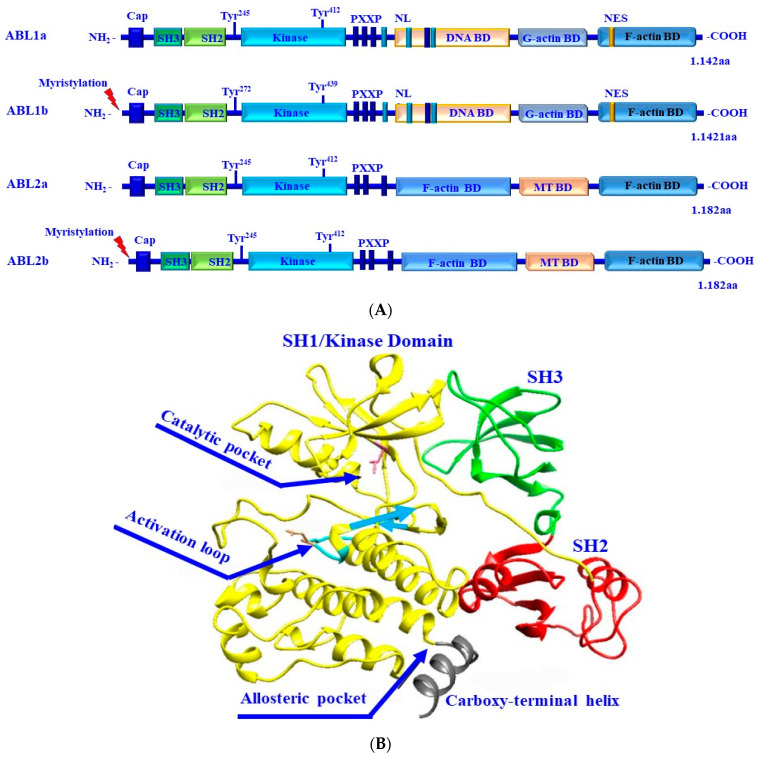
(**A**) Structural and functional domains of the Abl kinases. Alternative splicing of both Abl1 and Abl2 results in the production of protein variants/isoforms. The isoforms of Abl1 and Abl2, namely Abl1a and Abl2b, are characterized by their myristylation at the N-termini. Abl1 possesses two splice variants, Abl1a and Abl1b, and Abl2 also possesses two splice variants, Abl2a and Abl2b. The N-terminal region of Abl constitutes SH3, SH2, and SH1/kinase domains. The C- terminus of Abl constitutes conserved filamentous (F)-actin-binding (BD), globular (G)-actin-binding, and DNA-binding domains, and three nuclear localization signal (NLS) motifs and one nuclear export signal (NES). Both Abl1 and Abl2 kinases share the same domains in their N-terminus and are different in terms of their C-terminus. The C-terminus of Abl2 constitutes two BD domains and a microtubule (MT)-binding domain. Both Abl1 and Abl2 have conserved PXXP motifs to facilitate protein–protein interactions and tyrosine phosphorylation sites as indicated. (**B**) Three-dimensional conformation of Abl kinase protein. The different domains of the ABL kinase including SH3 (Green), SH2 (red), and SH1/kinase (yellow) domains, and the catalytic pocket, allosteric pocket, activation loop, and the carboxy-terminal helix are demonstrated.

**Figure 10 cancers-16-02754-f010:**
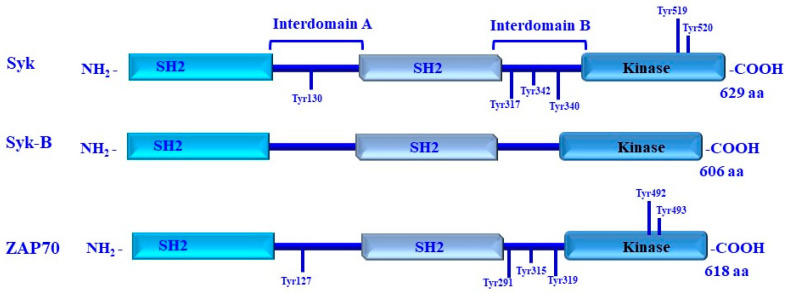
Structure of spleen tyrosine kinase (Syk)/Syk-B and ZAP-70 family protein tyrosine kinases. Syk kinases is organized in three functional domains, including two Src-homology 2 (SH2) domains and kinase domains. The three functional domains of the Syk kinase are connected via two A and B intermediate domains. Tyrosine residues of Syk that undergo phosphorylation are indicated. These tyrosine residues are essential for the regulation of enzymatic activity and to recruit other signaling proteins. The phosphorylation of Tyr^342^ and Tyr^346^ is dependent on the Src family kinases (SFK); phosphorylation of Tyr^130^, Tyr^317^, Tyr^519^, and Tyr^520^ is dependent on Syk itself and is target for autophosphorylation.

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
