# Peer review of "Non-Receptor Tyrosine Kinases: Their Structure and Mechanistic Role in Tumor Progression and Resistance"

_cancers, 2024, doi:10.3390/cancers16152754_

Round 1
Reviewer 1 Report
Comments and Suggestions for Authors
This review article, titled "Non-Receptor Tyrosine Kinases: Their Mechanistic Role in Tumor Progression and Treatment Resistance," explores the structure, function, and regulatory mechanisms of non-receptor tyrosine kinases (NRTKs) in cancer. The authors provide an in-depth look at the domain architecture of major NRTK families, including Ack, JAK, Fes, FAK, Tec, Src, Csk, Abl, and Syk, and discuss how aberrant activation of these kinases through gene amplification, mutation, and other means promotes tumorigenesis, metastasis, and resistance to targeted therapies. The potential of NRTKs as therapeutic targets in various cancer types is also examined.
Further, the article delves into the mechanisms by which NRTKs influence cellular processes such as growth factor signaling, cell cycle progression, cell survival, motility, and angiogenesis.
It highlights the therapeutic potential of targeting NRTKs, discussing various inhibitors and their effects on different cancers. The manuscript is well-structured, with logical transitions between sections and a smooth flow of ideas. It is extensively referenced, citing over 300 papers from the literature. Figures are used effectively to convey key concepts.
Minor Suggestions:
1. The manuscript sometimes uses abbreviations without defining them initially, which can confuse readers unfamiliar with the terms. For example, the acronym "RSKs" for receptor serine/threonine kinases is used before it is defined.
2. Introduction (Lines 54-73): The introduction provides a good foundation but could be more concise. Reducing redundancy in explaining the classification of PTKs would help in maintaining the reader's interest.
Overall Evaluation:
The manuscript provides a thorough and insightful review of non-receptor tyrosine kinases and their roles in tumor progression and treatment resistance. With some revisions to enhance clarity, it can serve as a valuable resource for researchers and clinicians in the field of cancer biology and therapeutics.
Author Response
Comment Authors’response to the comment of the Reviewer 1
Comments and Suggestions for Authors
This review article, titled "Non-Receptor Tyrosine Kinases: Their Mechanistic Role in Tumor Progression and Treatment Resistance," explores the structure, function, and regulatory mechanisms of non-receptor tyrosine kinases (NRTKs) in cancer. The authors provide an in-depth look at the domain architecture of major NRTK families, including Ack, JAK, Fes, FAK, Tec, Src, Csk, Abl, and Syk, and discuss how aberrant activation of these kinases through gene amplification, mutation, and other means promotes tumorigenesis, metastasis, and resistance to targeted therapies. The potential of NRTKs as therapeutic targets in various cancer types is also examined.
Further, the article delves into the mechanisms by which NRTKs influence cellular processes such as growth factor signaling, cell cycle progression, cell survival, motility, and angiogenesis.
It highlights the therapeutic potential of targeting NRTKs, discussing various inhibitors and their effects on different cancers. The manuscript is well-structured, with logical transitions between sections and a smooth flow of ideas. It is extensively referenced, citing over 300 papers from the literature. Figures are used effectively to convey key concepts.
Minor Suggestions:
Comment 1. The manuscript sometimes uses abbreviations without defining them initially, which can confuse readers unfamiliar with the terms. For example, the acronym "RSKs" for receptor serine/threonine kinases is used before it is defined.
Authors ‘response: Thank you very much for valuable comment. Accordingly, we defined all abbreviations overall in the main text of the manuscript.
See lines: 73-76; the following sentence [ NRTKs are both cytoplasmic and nuclear proteins, the localization of the cytoplasmic NRTK proteins are either free or anchored to the inner side of the cell membrane [15, 16]. The main function of the cytoplasmic NRTK protein is to mediate intracellular signals resulting from extracellular receptors-dependent activation [15, 17].].
See line:93; ATPs is replaced by adenosine triphosphate (ATP).
See lines: 128-129; Cdc42/Rac-interactive domain (CRIB).
See line: 136-137; guanosine triphosphatase (GTPase).
See line: 425-426; breast tumor kinase (BRK) family kinases (BFKs)
Comment 2. Introduction (Lines 54-73): The introduction provides a good foundation but could be more concise. Reducing redundancy in explaining the classification of PTKs would help in maintaining the reader's interest.
Authors ‘response: Thank you very much for your valuable comment. Accordingly, we modified the introduction as required.
See lines: 51-67; the following Text [Protein tyrosine kinases (PTKs) are a large multigene family present in most eukaryotic organisms in addition to playing important roles in cellular processes such as metabolism, migration, survival, proliferation and differentiation [1-3]. Uncontrolled activation of PTKs can cause various diseases, including cancer, diabetes and atherosclerosis [3,4] Thus, in addition to being key signaling molecules, PTKs are recognized as important drug targets in many diseases, particularly cancers [1, 2]. Due to the great diversity within the PTK family and their important role in cellular processes, the field of drug development has focused on the use of PTKs as target molecules [2,3]. PTKs contain an evolutionarily conserved catalytic kinase domain with the ability to phosphorylate substrate proteins at tyrosine residues and can be divided into two classes depending on the presence of a transmembrane (TM) domain: receptor tyrosine kinases (RTKs) and non-receptor tyrosine kinases (NRTKs) [4,5]. PTKs -mediated phosphorylation of serine, threonine, or tyrosine residues rapidly and reversibly alters protein function, conformation, and interaction to drive cellular functioning under normal and pathological conditions [3,4]. PTKs are classified into two groups, tyrosine kinases and serine/ threonine kinases [6]. Receptor serine/threonine kinases (RSTKs) are transmembrane proteins characterized by extracellular ligand‐binding domains and cytoplasmic kinase domains which act as signaling receptors for members of transforming growth factor β to maintain the progression of signal transduction processes [7, 8]. ]
Overall Evaluation: The manuscript provides a thorough and insightful review of non-receptor tyrosine kinases and their roles in tumor progression and treatment resistance. With some revisions to enhance clarity, it can serve as a valuable resource for researchers and clinicians in the field of cancer biology and therapeutics.
Authors ‘response: Thank you very much for the valuable comment

Reviewer 2 Report
Comments and Suggestions for Authors
In this manuscript entitled "Non-receptor tyrosine kinases: Their Mechanistic Role in Tumour Progression and Treatment Resistance", Hassan et al. review this important group of protein kinases and signaling pathways involved in carcinogenesis. The latter with the aim of considering them as therapeutic targets, according to the work reported in the last decades, as well as the evidence of drugs in clinical use whose pharmacological targets are this class of proteins. Therefore, it can be a very useful material for researchers in the field of medicinal chemistry focused on the development of new chemotherapeutic agents.
Although a great deal of information is presented on the structural aspects of these NRTKs, this manuscript has serious shortcomings that make it less useful as a reference material and therefore, in its current form, it is not recommended for publication.
Therefore, with all due respect to the work done, the following are some of the most important points to be considered for a new, corrected and improved version:
Major revisions
1. From the current title the role of these kinases in tumor progression will be addressed, which is not clear throughout the manuscript because most of the information is focused on structural aspects of each protein belonging to the NRTK families, their composition and how certain alterations of these are responsible for carcinogenic processes only in some cases. The problem of the title is much more complex when it comes to resistance to treatment, because this is a point that is not addressed in the manuscript, even though there is ample evidence of cases in which the possible mechanisms for the drugs used are related to mutations of these kinases, as is the case for imatinib, ibrutinib, among others. Furthermore, this point is mentioned in the conclusions and is not adequately supported throughout the manuscript.
2. Although many of the original papers related to the initial studies of these kinases are cited, as this is a review based on the current situation, it would be expected that the percentage of cited papers would be higher for the years 2020-2024 and not only about 20% of the total literature.
3. There is a lack of diagrams or figures that help to integrate the content related to the role of these kinases with signaling pathways related to cancer, and this considering that this idea is more developed only for some families, such as Ack, and is very superficial for others, such as Tec. This is particularly the case given the known role of BTK in hematological cancers and other solid tumors.
4. I think it would have been more appropriate to deal with each family of NRTKs from a structural point of view and their role in the processes of carcinogenesis directly, rather than separately as presented, because what is presented in points 3 and 4 disorganizes the information much more and makes it repetitive in some cases. The latter is more evident in the paragraphs between lines 519-559. Here, unfortunately, the Bcr-Abl is very poorly written, the information is duplicated, and the writing is somewhat messy.
5. Finally, it is worth mentioning that, although it is necessary to know the structure of an NRTK in terms of its domain composition and homology with other proteins, since this evidence is directly correlated with the cellular mechanisms that generate cancer and its possible impact on the treatment of this pathology, it is something that this work fails to appreciate.
Minor revisions:
1. Correctly name the genes mentioned throughout the manuscript and the corresponding proteins expressed by them. Italicize the genes and use the correct nomenclature.
2. Use italics for species names. For example: Drosophila melanogaster, etc.
3. Standardize the way figures are cited in the text, for example: Figure 1, Figure 2. and if they are cited: ....are outlined as shown (Figure 9), it should be: ...are outlined as shown in Figure 9.
4. Line 427 refers to Figure 9 when it should be Figure 8.
Author Response
Authors ‘response to the comment of the reviewer 2
As required enclosed find please our response point-for-point to the valuable comment of the reviewer 02
Comment of the Reviewer 02
Comments and Suggestions for Authors
In this manuscript entitled "Non-receptor tyrosine kinases: Their Mechanistic Role in Tumour Progression and Treatment Resistance", Hassan et al. review this important group of protein kinases and signaling pathways involved in carcinogenesis. The latter with the aim of considering them as therapeutic targets, according to the work reported in the last decades, as well as the evidence of drugs in clinical use whose pharmacological targets are this class of proteins. Therefore, it can be a very useful material for researchers in the field of medicinal chemistry focused on the development of new chemotherapeutic agents.
Although a great deal of information is presented on the structural aspects of these NRTKs, this manuscript has serious shortcomings that make it less useful as a reference material and therefore, in its current form, it is not recommended for publication.
Therefore, with all due respect to the work done, the following are some of the most important points to be considered for a new, corrected and improved version:
Authors ’response: thank you very much for your valuable comment
Major revisions
Comment1. From the current title the role of these kinases in tumor progression will be addressed, which is not clear throughout the manuscript because most of the information is focused on structural aspects of each protein belonging to the NRTK families, their composition and how certain alterations of these are responsible for carcinogenic processes only in some cases. The problem of the title is much more complex when it comes to resistance to treatment, because this is a point that is not addressed in the manuscript, even though there is ample evidence of cases in which the possible mechanisms for the drugs used are related to mutations of these kinases, as is the case for imatinib, ibrutinib, among others. Furthermore, this point is mentioned in the conclusions and is not adequately supported throughout the manuscript.
Authors ‘response: Thank you very much for your valuable comment. Accordingly, we modified the title to be more suitable as required. In addition, we added more information about the reliability of the members of NRTKs as therapeutic target of many inhibitors, that have been approved as anticancer agents.
See Lines: 2-3; the current title [ Non-Receptor tyrosine kinases: Their Mechanistic Role in Tu-mor Progression and Treatment Resistance] has been modified and replaced by the following one [Non-Receptor tyrosine kinases: Their Structure and Mechanistic Role in Tumor Progression and Resistance]
See Lines: 756-793 ; the following text [Several kinase inhibitors targeting family members have been further developed and approved for their clinical application. The most common clinically applied inhibitors include Imatinib/(STI-571), Dasatinib/(BMS-354825, Nilotinib/(AMN107), Bosutinib (SKI-606), Radotinib6/(IY-5511), Ponatinib/(AP24534), Asciminib/(ABL001), Ibrutinib/(PCI-32765), Acalabrutinib/(ACP-196), Zanubrutinib/(BGB-3111), Ruxolitinib/(INC424), and Fedratinib/(SAR302503, TG101348). The Imatinib is an ATP competitive type II tyrosine kinase inhibitor (TKI) [340]. Thus, a mutation in the ATP-interacting gatekeeper residue threonine 315 (T315) of the ATP binding pocket is critical for the accessibility of the ATP binding pocket and leads to maintenance of the active conformation of ABL and resistance to imatinib and related TKIs [340]. However, the development of a second generation BCR-ABL inhibitors such as Nilotinib/(AMN107), whose clinical utilization confirmed the therapeutic potential of second generation TKIs to overcome imatinib resistance caused by ABL kinase point mutations [340]. Of note Nilotinib is an ATP-competitive type II kinase inhibitor with greatly improved potency compared to imatinib [340, 341].
Bosutinib is an ATP competitive dual SFK/ABL inhibitor in addition its inhibitory effects against mutated or amplified BCR-ABL associated with imatinib resistance [340-343]. Also, the inhibitory effect of bosutinib has been demonstrated in patients with Philadelphia chromosome-positive chronic myeloid leukemia (Ph+ CML), particularly who are resistant to the treatment with imatinib [344]. Other agents such as radotinib that has been cliniclly approved as second-generation BCR-ABL inhibitor based on its ability to exhibit inhibitory effects on wild-type and some imatinib resistant mutant forms of BCR-ABL [345]. Also, the allosteric inhibitor asciminib is characterized by its affinity to bind to the myristate pocket of BCRABL and is effective against T315I-mutant BCR-ABL [346]. The third-generation inhibitor, Ponatinib, is clinically approved for the treatment of both wild-type and T315I-mutant BCR-ABL [340, 347]. Also, ponatinib displayed inhibitory effects on the activity of multiple kinases, including both FLT3, , and Src [347]. As is known, the Src-family kinases (SFK: Blk, Fgr, Frk, Fyn, Hck, Lck, Lyn, Src, Yes, and Yrk) contain a conserved domain organization consisting of a myristoylated N-terminal segment (SH4 domain), followed by SH3, SH2, linker, and tyrosine kinase domains and a short C-terminal tail [348]. Like ABL, SFKs take over an inactive conformation through autoinhibitory intramolecular interactions involving phosphorylation at Y527/Y530 [349]. Thus, the dephosphorylation of Y527/ Y530 causes destabilization of intramolecular interactions, leading to SFK activation by interaction with RTKs, G protein-coupled receptors, and focal adhesion kinase via its SH2 or SH3 domains and subsequent autophosphorylation at Y416/Y419 [349]. In addition to their crucial role in the regulation of in cell proliferation, adhesion, migration, invasion, metastasis, angiogenesis, and therapeutic resistance in cancer activated SFKs can act as key nodes of multiple oncogenic signal transduction pathways [ 349, 350]. Thus, based on its functional role in tumor progression SFK is a potential target for efficacious anticancer therapeutic regimens [350]. The potential of Dasatinib to target multiple kinases, including Src, Fgr, Fyn, Hck, Lck, Lyn, and Yes, [340, 349, 350].] has been added to the section of the NRTKs as therapeutic target in Tumor treatment.
References Section
See Lines: 1646-1672; the following references:
- Rossari, F.; Minutolo, F.; Orciuolo, E. Past, present, and future of Bcr-Abl inhibitors: from chemical development to clinical efficacy. J Hematol Oncol 2018, 11(1), 84. DOI: 10.1186/s13045-018-0624-2.
- Giles, F. J.; O'Dwyer, M.; Swords, R. Class effects of tyrosine kinase inhibitors in the treatment of chronic myeloid leukemia. Leukemia 2009, 23(10), 1698-1707. DOI: 10.1038/leu.2009.111. ].
- Weisberg, E.; Manley, P.; Mestan, J.; Cowan-Jacob, S.; Ray, A.; Griffin, J. D. AMN107 (nilotinib): a novel and selective inhibitor of BCR-ABL. Br J Cancer 2006, 94(12), 1765-1769. DOI: 10.1038/sj.bjc.6603170.
- Puttini, M.; Coluccia, A. M.; Boschelli, F.; Cleris, L.; Marchesi, E.; Donella-Deana, A.; Ahmed, S.; Redaelli, S.; Piazza, R.; Magistroni, V.; et al. In vitro and in vivo activity of SKI-606, a novel Src-Abl inhibitor, against imatinib-resistant Bcr-Abl+ neoplastic cells. Cancer Res 2006, 66(23), 11314-11322. DOI: 10.1158/0008-5472.CAN-06-1199.
- Hochhaus, A.; Saussele, S.; Rosti, G.; Mahon, F. X.; Janssen, J. J. W. M.; Hjorth-Hansen, H.; Richter, J.; Buske, C.; Committee, E. G. Chronic myeloid leukaemia: ESMO Clinical Practice Guidelines for diagnosis, treatment and follow-up. Ann Oncol 2017, 28(suppl_4), iv41-iv51. DOI: 10.1093/annonc/mdx219.].
- Eskazan, A. E.; Keskin, D. Radotinib and its clinical potential in chronic-phase chronic myeloid leukemia patients: an update. Ther Adv Hematol 2017, 8(9), 237-243. DOI: 10.1177/2040620717719851
- Hughes, T. P.; Mauro, M. J.; Cortes, J. E.; Minami, H.; Rea, D.; DeAngelo, D. J.; Breccia, M.; Goh, Y. T.; Talpaz, M.; Hochhaus, A.; et al. Asciminib in Chronic Myeloid Leukemia after ABL Kinase Inhibitor Failure. N Engl J Med 2019, 381(24), 2315-2326. DOI: 10.1056/NEJMoa1902328.
- Tan, F. H.; Putoczki, T. L.; Stylli, S. S.; Luwor, R. B. Ponatinib: a novel multi-tyrosine kinase inhibitor against human malignancies. Onco Targets Ther 2019, 12, 635-645. DOI: 10.2147/OTT.S189391.
- Wheeler, D. L.; Iida, M.; Dunn, E. F. The role of Src in solid tumors. Oncologist 2009, 14(7), 667-678. DOI: 10.1634/theoncologist.2009-0009.
- Zhang, S.; Yu, D. Targeting Src family kinases in anti-cancer therapies: turning promise into triumph. Trends Pharmacol Sci 2012, 33(3), 122-128. DOI: 10.1016/j.tips.2011.11.00.
- Das, J.; Chen, P.; Norris, D.; Padmanabha, R.; Lin, J.; Moquin, R. V.; Shen, Z.; Cook, L. S.; Doweyko, A. M.; Pitt, S.; et al. 2-aminothiazole as a novel kinase inhibitor template. Structure-activity relationship studies toward the discovery of N-(2-chloro-6-methylphenyl)-2-[[6-[4-(2-hydroxyethyl)-1- piperazinyl)]-2-methyl-4-pyrimidinyl]amino)]-1,3-thiazole-5-carboxamide (dasatinib, BMS-354825) as a potent pan-Src kinase inhibitor. J Med Chem 2006, 49(23), 6819-6832. DOI: 10.1021/jm060727j.
Comment 2. Although many of the original papers related to the initial studies of these kinases are cited, as this is a review based on the current situation, it would be expected that the percentage of cited papers would be higher for the years 2020-2024 and not only about 20% of the total literature.
Authors ‘response: Thank you very much for your comment. Accordingly, we updated the references as required.
Comment 3. There is a lack of diagrams or figures that help to integrate the content related to the role of these kinases with signaling pathways related to cancer, and this considering that this idea is more developed only for some families, such as Ack, and is very superficial for others, such as Tec. This is particularly the case given the known role of BTK in hematological cancers and other solid tumors.
Authors’ response: Thank you very much for valuable comment. Accordingly, we extended the text with focus on the Src-FAK-dependent pathways, as an example for NRTKs-mediated signaling pathways in normal and cancer cells. Also, we included a figure to make it easier for the reader to follow.
Main Text
See Lines:6481-685, the following paragraphs [The interaction of Src with the activated RTKs leads to the formation of a positive regulatory loop that contributes to the resilience and persistence of RTK signaling. The binding of RTK to its respective ligands leads to receptor dimerization and auto transphosphorylation at tyrosine residues of the cytoplasmic domain; the resulting phospho-tyrosine acts as a docking site to recruit and activate Src, which then phosphorylates RTK and increases RTK tyrosine kinase activity, and to create SH2-binding recruitment sites [ 243, 244].
As consequence, activated Src allows the binding of Grb2-Sos complex leading to activation of Ras/MAPK, and the activation of PI3K resulting in the activation of Akt [245, 246].
Integrins are transmembrane adhesion receptors localized at cell-matrix contact sites, where they connect components of the extracellular matrix (ECM) to the actin cytoskeleton and interact with several structural and signaling molecules, including talin, paxillin, vinculin, a-actinin, FAK, and Src [247, 248].
These important downstream mediators, particularly FAK and Src, of integrins with which they interact either directly or indirectly to induce adhesion-dependent responses. These important downstream mediators, particularly FAK and Src, of integrins with which they interact either directly or indirectly to enhance adhesion dependent responses [249, 250]. Thus, the cytoplasmic tail of ß3 integrins can directly interact with the SH3 domain of Src upon ECM ligand binding and promote its activation; activated Src can in turn promote the activation of the guanine nucleotide exchange factor (GEF) Vav1 and T-cell lymphoma invasion and metastasis 1 (Tiam1), which are responsible for Rac activation, and thus induce the stimulation of actin-driven positive activity at the site of integrin binding [249-251]. The interaction between the ß3 integrin and the SH3 domain of Src might underlie the stimulation of STAT3 and FAK signaling to promote tumor growth. Phosphorylation of the ß1 integrin might be required to inhibit Rho-mediated cytoskeletal contractility and thus be involved in the generation of the transformed phenotype. [252, 253]. An important function of Src appears to be to weaken the connections between ECM, integrins and cytoskeleton and to induce adhesion turnover and remodeling of the actin cytoskeleton [252, 253 ], and these functions imply the role of another important mediator of integrin signaling, such as FAK. The tyrosine kinase FAK acts both as a signaling molecule and a scaffold able to recruit Src and the Src substrates to sites of integrin engagement, and has an important role in cell cycle progression and survival as well as in adhesion and migration [254, 255]. In addition to ECM integrins, soluble growth factors also promote FAK activation. As a result, FAK integrates signals from the growth factor receptor and the matrix to trigger biological responses, including the induction of a migratory phenotype [253, 257 ].
The transient dimerization, intermolecular autophosphorylation and activation of FAK has been reported to result from cell-matrix contact-dependent clustering of integrins [258, 259 ]. To that end, NRTKs are involved in multiple signaling pathways that regulate vital functions such as cell proliferation and differentiation, and plays a role in human neoplasms, inflammatory and autoimmune diseases. ] have been added to the section of NRTKs-mediated pathways in normal and cancer cells.
References Section
See lines:1414-1449 ; the following references:
- Maruyama, I. N. Mechanisms of activation of receptor tyrosine kinases: monomers or dimers. Cells 2014, 3 (2), 304-330. DOI: 10.3390/cells3020304.
- Trenker, R.; Jura, N. Receptor tyrosine kinase activation: From the ligand perspective. Curr Opin Cell Biol 2020, 63, 174-185. DOI: 10.1016/j.ceb.2020.01.016.
- Raji, L.; Tetteh, A.; Amin, A. R. M. R. Role of c-Src in Carcinogenesis and Drug Resistance. Cancers (Basel) 2023, 16(1). DOI: 10.3390/cancers16010032.
- Penuel, E.; Martin, G. S. Transformation by v-Src: Ras-MAPK and PI3K-mTOR mediate parallel pathways. Mol Biol Cell 1999, 10 (6), 1693-1703. DOI: 10.1091/mbc.10.6.1693.
- Lu, F.; Zhu, L.; Bromberger, T.; Yang, J.; Yang, Q.; Liu, J.; Plow, E. F.; Moser, M.; Qin, J. Mechanism of integrin activation by talin and its cooperation with kindlin. Nat Commun 2022, 13(1), 2362. DOI: 10.1038/s41467-022-30117-w.
- Li, Z.; Lee, H.; Zhu, C. Molecular mechanisms of mechanotransduction in integrin-mediated cell-matrix adhesion. Exp Cell Res 2016, 349 (1), 85-94. DOI: 10.1016/j.yexcr.2016.10.001.
- Luo, J.; Zou, H.; Guo, Y.; Tong, T.; Ye, L.; Zhu, C.; Deng, L.; Wang, B.; Pan, Y.; Li, P. SRC kinase-mediated signaling pathways and targeted therapies in breast cancer. Breast Cancer Res 2022, 24 (1), 99. DOI: 10.1186/s13058-022-01596-y.
- Kazemein Jasemi, N. S.; Ahmadian, M. R. Allosteric regulation of GRB2 modulates RAS activation. Small GTPases 2022, 13 (1), 282-286. DOI: 10.1080/21541248.2022.2089001.
- Beadnell, T. C.; Nassar, K. W.; Rose, M. M.; Clark, E. G.; Danysh, B. P.; Hofmann, M. C.; Pozdeyev, N.; Schweppe, R. E. Src-mediated regulation of the PI3K pathway in advanced papillary and anaplastic thyroid cancer. Oncogenesis 2018, 7 (2), 23. DOI: 10.1038/s41389-017-0015-5.].
- Kanchanawong, P.; Calderwood, D. A. Organization, dynamics and mechanoregulation of integrin-mediated cell-ECM adhesions. Nat Rev Mol Cell Biol 2023, 24(2), 142-161. DOI: 10.1038/s41580-022-00531-5.
- Mitra, S. K.; Schlaepfer, D. D. Integrin-regulated FAK-Src signaling in normal and cancer cells. Curr Opin Cell Biol 2006, 18 (5), 516-523. DOI: 10.1016/j.ceb.2006.08.011.
- Li, S.; Sampson, C.; Liu, C.; Piao, H. L.; Liu, H. X. Integrin signaling in cancer: bidirectional mechanisms and therapeutic opportunities. Cell Commun Signal 2023, 21(1), 266. DOI: 10.1186/s12964-023-01264-4.
- Yousefi, H.; Vatanmakanian, M.; Mahdiannasser, M.; Mashouri, L.; Alahari, N. V.; Monjezi, M. R.; Ilbeigi, S.; Alahari, S. K. Understanding the role of integrins in breast cancer invasion, metastasis, angiogenesis, and drug resistance. Oncogene 2021, 40 (6), 1043-1063. DOI: 10.1038/s41388-020-01588-2.
- Huveneers, S.; Danen, E. H. Adhesion signaling - crosstalk between integrins, Src and Rho. J Cell Sci 2009, 122(Pt 8), 1059-1069. DOI: 10.1242/jcs.039446.
- Pang, X.; He, X.; Qiu, Z.; Zhang, H.; Xie, R.; Liu, Z.; Gu, Y.; Zhao, N.; Xiang, Q.; Cui, Y. Targeting integrin pathways: mechanisms and advances in therapy. Signal Transduct Target Ther 2023, 8(1), 1. DOI: 10.1038/s41392-022-01259-6.
- [ 275] Playford, M. P.; Schaller, M. D. The interplay between Src and integrins in normal and tumor biology. Oncogene 2004, 23(48), 7928-7946. DOI: 10.1038/sj.onc.1208080.
- Murphy, J. M.; Rodriguez, Y. A. R.; Jeong, K.; Ahn, E. E.; Lim, S. S. Targeting focal adhesion kinase in cancer cells and the tumor microenvironment. Exp Mol Med 2020, 52 (6), 877-886. DOI: 10.1038/s12276-020-0447-4. ] have been added to the section of references
Figures
See Figure 5B; The following figure: Figure 5B, has been created
Legends to Figures
See Lines: 314-347; the following legend [Fig.5B Src/FAK-mediated transduction pathways which contribute to cancer progression. Focal adhesion kinase (FAK)-mediated signaling cascades involved in tumor progression. FAK is activated by Integrins to enhance the activity of phosphoinositide 3-kinase PI3K)s These non-receptor tyrosine kinase-dependent pathways are associated with the stimulation of cell proliferation, angiogenesis and survival. Phosphorylated FAK can recruit Grb2 and the p85 regulatory subunit of PI3K, thus leading respectively to stimulation of Grb2/Ras/MAPK and PI3K/Akt pathways which enhance survival and proliferation, ] has been added to the main Text of the manuscript
Comment 4. I think it would have been more appropriate to deal with each family of NRTKs from a structural point of view and their role in the processes of carcinogenesis directly, rather than separately as presented, because what is presented in points 3 and 4 disorganizes the information much more and makes it repetitive in some cases. The latter is more evident in the paragraphs between lines 519-559. Here, unfortunately, the Bcr-Abl is very poorly written, the information is duplicated, and the writing is somewhat messy.
Authors ‘response: Thank you very much for your comment. Accordingly, we added more information about Bcr-Abl
Main Text
See Lines: 484-497; the flowing paragraph [The activation of ABL family members (e.g. Abl1, ARG) by different mechanisms including, mutation, dimerization, phosphorylation, or binding proteins that disrupt intramolecular interactions [200-202], and by the phosphorylation within kinase and interlinked regions (Y412, Y245) [203-206]. However, the disruption of autoinhibition in response to the translocation of Abl1 or Abl2 next to a variety of different genes (e.g., BCR, Tel, ETV6), causes constitutive activation leading to the development different tumor types including leukemia [207]. Although both c-Abl and Arg are involved in the development of a variety of human leukemias, mutations and/or activating translocations of these kinases so far are not identified in solid tumors [208]. Therefore, it is expected that c-Abl and Arg are not activated in other cancers. However, data from the immune histochemistry analysis of c-Abl and/or Arg showed that they are overexpressed in some solid tumors, including brain, lung, ovarian, colon and prostate cancer, as well as in chondrosarcomas, liposarcomas, diffuse gastric adenocarcinomas, oral squamous cell carcinomas, atypical teratoid and malignant rhabdoid tumors and endometrial carcinomas, compared to normal tissue or benign tumors. In addition, c-Abl amplification was found in renal medulla carcinomas [209-211]. The structure and functional domains, as well as the 3D- of ABL kinase protein are outlined in figure 9 A and B.] has been added to the section NRTKs-mediated pathways in normal and cancer cells.
References
See lines: 1315-1343; The following references:
- Aoyama, K.; Yuki, R.; Horiike, Y.; Kubota, S.; Yamaguchi, N.; Morii, M.; Ishibashi, K.; Nakayama, Y.; Kuga, T.; Hashimoto, Y.; et al. Formation of long and winding nuclear F-actin bundles by nuclear c-Abl tyrosine kinase. Exp Cell Res 2013, 319(20), 3251-3268. DOI: 10.1016/j.yexcr.2013.09.003.
- Hu, H.; Bliss, J. M.; Wang, Y.; Colicelli, J. RIN1 is an ABL tyrosine kinase activator and a regulator of epithelial-cell adhesion and migration. Curr Biol 2005, 15(9), 815-823. DOI: 10.1016/j.cub.2005.03.049.
- Sriram, G.; Reichman, C.; Tunceroglu, A.; Kaushal, N.; Saleh, T.; Machida, K.; Mayer, B.; Ge, Q.; Li, J.; Hornbeck, P.; et al. Phosphorylation of Crk on tyrosine 251 in the RT loop of the SH3C domain promotes Abl kinase transactivation. Oncogene 2011, 30(46), 4645-4655. DOI: 10.1038/onc.2011.170.
- Cao, X.; Tanis, K. Q.; Koleske, A. J.; Colicelli, J. Enhancement of ABL kinase catalytic efficiency by a direct binding regulator is independent of other regulatory mechanisms. J Biol Chem 2008, 283(46), 31401-31407. DOI: 10.1074/jbc.M804002200.
- Plattner, R.; Kadlec, L.; DeMali, K. A.; Kazlauskas, A.; Pendergast, A. M. c-Abl is activated by growth factors and Src family kinases and has a role in the cellular response to PDGF. Genes Dev 1999, 13(18), 2400-2411. DOI: 10.1101/gad.13.18.2400.
- Brasher, B. B.; Van Etten, R. A. c-Abl has high intrinsic tyrosine kinase activity that is stimulated by mutation of the Src homology 3 domain and by autophosphorylation at two distinct regulatory tyrosines. J Biol Chem 2000, 275(45), 35631-35637. DOI: 10.1074/jbc.M005401200.
- Smith, K. M.; Van Etten, R. A. Activation of c-Abl kinase activity and transformation by a chemical inducer of dimerization. J Biol Chem 2001, 276(26), 24372-24379. DOI: 10.1074/jbc.M100786200.
- Fan, P. D.; Cong, F.; Goff, S. P. Homo- and hetero-oligomerization of the c-Abl kinase and Abelson-interactor-1. Cancer Res 2003, 63(4), 873-877
- Sawyers, C. L.; McLaughlin, J.; Goga, A.; Havlik, M.; Witte, O. The nuclear tyrosine kinase c-Abl negatively regulates cell growth. Cell 1994, 77(1), 121-131. DOI: 10.1016/0092-8674(94)90240-2.
- Greuber, E. K.; Smith-Pearson, P.; Wang, J.; Pendergast, A. M. Role of ABL family kinases in cancer: from leukaemia to solid tumours. Nat Rev Cancer 2013, 13(8), 559-571. DOI: 10.1038/nrc3563.
- Ganguly, S. S.; Fiore, L. S.; Sims, J. T.; Friend, J. W.; Srinivasan, D.; Thacker, M. A.; Cibull, M. L.; Wang, C.; Novak, M.; Kaetzel, D. M.; et al. c-Abl and Arg are activated in human primary melanomas, promote melanoma cell invasion via distinct pathways, and drive metastatic progression. Oncogene 2012, 31(14), 1804-1816. DOI: 10.1038/onc.2011.361.
- Koos, B.; Jeibmann, A.; Lünenbürger, H.; Mertsch, S.; Nupponen, N. N.; Roselli, A.; Leuschner, I.; Paulus, W.; Frühwald, M. C.; Hasselblatt, M. The tyrosine kinase c-Abl promotes proliferation and is expressed in atypical teratoid and malignant rhabdoid tumors. Cancer 2010, 116(21), 5075-5081. DOI: 10.1002/cncr.25420. ] has been added to the section of the references
.
- Finally, it is worth mentioning that, although it is necessary to know the structure of an NRTK in terms of its domain composition and homology with other proteins, since this evidence is directly correlated with the cellular mechanisms that generate cancer and its possible impact on the treatment of this pathology, it is something that this work fails to appreciate.
Authors ‘response: Thank you very much for your comment. Accordingly, we added mor information about The impact of NRTKs as therapeutic target.
Main Text
See Lines: 756--793; the following paragraphs [ Several kinase inhibitors targeting family members have been further developed and approved for their clinical application. The most common clinically applied inhibitors include Imatinib/(STI-571), Dasatinib/(BMS-354825, Nilotinib/(AMN107), Bosutinib (SKI-606), Radotinib6/(IY-5511), Ponatinib/(AP24534), Asciminib/(ABL001), Ibrutinib/(PCI-32765), Acalabrutinib/(ACP-196), Zanubrutinib/(BGB-3111), Ruxolitinib/(INC424), and Fedratinib/(SAR302503, TG101348). The Imatinib is an ATP competitive type II tyrosine kinase inhibitor (TKI) [340]. Thus, a mutation in the ATP-interacting gatekeeper residue threonine 315 (T315) of the ATP binding pocket is critical for the accessibility of the ATP binding pocket and leads to maintenance of the active conformation of ABL and resistance to imatinib and related TKIs [340]. However, the development of a second generation BCR-ABL inhibitors such as Nilotinib/(AMN107), whose clinical utilization confirmed the therapeutic potential of second generation TKIs to overcome imatinib resistance caused by ABL kinase point mutations [340]. Of note Nilotinib is an ATP-competitive type II kinase inhibitor with greatly improved potency compared to imatinib [340, 341].
Bosutinib is an ATP competitive dual SFK/ABL inhibitor in addition its inhibitory effects against mutated or amplified BCR-ABL associated with imatinib resistance [340-343]. Also, the inhibitory effect of bosutinib has been demonstrated in patients with Philadelphia chromosome-positive chronic myeloid leukemia (Ph+ CML), particularly who are resistant to the treatment with imatinib [344]. Other agents such as radotinib that has been cliniclly approved as second-generation BCR-ABL inhibitor based on its ability to exhibit inhibitory effects on wild-type and some imatinib resistant mutant forms of BCR-ABL [345]. Also, the allosteric inhibitor asciminib is characterized by its affinity to bind to the myristate pocket of BCRABL and is effective against T315I-mutant BCR-ABL [346]. The third-generation inhibitor, Ponatinib, is clinically approved for the treatment of both wild-type and T315I-mutant BCR-ABL [340, 347]. Also, ponatinib displayed inhibitory effects on the activity of multiple kinases, including both FLT3, , and Src [347]. As is known, the Src-family kinases (SFK: Blk, Fgr, Frk, Fyn, Hck, Lck, Lyn, Src, Yes, and Yrk) contain a conserved domain organization consisting of a myristoylated N-terminal segment (SH4 domain), followed by SH3, SH2, linker, and tyrosine kinase domains and a short C-terminal tail [348]. Like ABL, SFKs take over an inactive conformation through autoinhibitory intramolecular interactions involving phosphorylation at Y527/Y530 [349]. Thus, the dephosphorylation of Y527/ Y530 causes destabilization of intramolecular interactions, leading to SFK activation by interaction with RTKs, G protein-coupled receptors, and focal adhesion kinase via its SH2 or SH3 domains and subsequent autophosphorylation at Y416/Y419 [349]. In addition to their crucial role in the regulation of in cell proliferation, adhesion, migration, invasion, metastasis, angiogenesis, and therapeutic resistance in cancer activated SFKs can act as key nodes of multiple oncogenic signal transduction pathways [ 349, 350]. Thus, based on its functional role in tumor progression SFK is a potential target for efficacious anticancer therapeutic regimens [350]. The potential of Dasatinib to target multiple kinases, including Src, Fgr, Fyn, Hck, Lck, Lyn, and Yes, [340, 349, 350]. ] have been added to the section of NRTK as therapeutic target.
References
See lines:1646-1672; the following references:
- Rossari, F.; Minutolo, F.; Orciuolo, E. Past, present, and future of Bcr-Abl inhibitors: from chemical development to clinical efficacy. J Hematol Oncol 2018, 11 (1), 84. DOI: 10.1186/s13045-018-0624-2.
- Giles, F. J.; O'Dwyer, M.; Swords, R. Class effects of tyrosine kinase inhibitors in the treatment of chronic myeloid leukemia. Leukemia 2009, 23 (10), 1698-1707. DOI: 10.1038/leu.2009.111. ].
- Weisberg, E.; Manley, P.; Mestan, J.; Cowan-Jacob, S.; Ray, A.; Griffin, J. D. AMN107 (nilotinib): a novel and selective inhibitor of BCR-ABL. Br J Cancer 2006, 94 (12), 1765-1769. DOI: 10.1038/sj.bjc.6603170.
- Puttini, M.; Coluccia, A. M.; Boschelli, F.; Cleris, L.; Marchesi, E.; Donella-Deana, A.; Ahmed, S.; Redaelli, S.; Piazza, R.; Magistroni, V.; et al. In vitro and in vivo activity of SKI-606, a novel Src-Abl inhibitor, against imatinib-resistant Bcr-Abl+ neoplastic cells. Cancer Res 2006, 66 (23), 11314-11322. DOI: 10.1158/0008-5472.CAN-06-1199.
- Hochhaus, A.; Saussele, S.; Rosti, G.; Mahon, F. X.; Janssen, J. J. W. M.; Hjorth-Hansen, H.; Richter, J.; Buske, C.; Committee, E. G. Chronic myeloid leukaemia: ESMO Clinical Practice Guidelines for diagnosis, treatment and follow-up. Ann Oncol 2017, 28 (suppl_4), iv41-iv51. DOI: 10.1093/annonc/mdx219.].
- Eskazan, A. E.; Keskin, D. Radotinib and its clinical potential in chronic-phase chronic myeloid leukemia patients: an update. Ther Adv Hematol 2017, 8 (9), 237-243. DOI: 10.1177/2040620717719851
- Hughes, T. P.; Mauro, M. J.; Cortes, J. E.; Minami, H.; Rea, D.; DeAngelo, D. J.; Breccia, M.; Goh, Y. T.; Talpaz, M.; Hochhaus, A.; et al. Asciminib in Chronic Myeloid Leukemia after ABL Kinase Inhibitor Failure. N Engl J Med 2019, 381 (24), 2315-2326. DOI: 10.1056/NEJMoa1902328.
- Tan, F. H.; Putoczki, T. L.; Stylli, S. S.; Luwor, R. B. Ponatinib: a novel multi-tyrosine kinase inhibitor against human malignancies. Onco Targets Ther 2019, 12, 635-645. DOI: 10.2147/OTT.S189391.
- Wheeler, D. L.; Iida, M.; Dunn, E. F. The role of Src in solid tumors. Oncologist 2009, 14 (7), 667-678. DOI: 10.1634/theoncologist.2009-0009.
- Zhang, S.; Yu, D. Targeting Src family kinases in anti-cancer therapies: turning promise into triumph. Trends Pharmacol Sci 2012, 33 (3), 122-128. DOI: 10.1016/j.tips.2011.11.00.
- Das, J.; Chen, P.; Norris, D.; Padmanabha, R.; Lin, J.; Moquin, R. V.; Shen, Z.; Cook, L. S.; Doweyko, A. M.; Pitt, S.; et al. 2-aminothiazole as a novel kinase inhibitor template. Structure-activity relationship studies toward the discovery of N-(2-chloro-6-methylphenyl)-2-[[6-[4-(2-hydroxyethyl)-1- piperazinyl)]-2-methyl-4-pyrimidinyl]amino)]-1,3-thiazole-5-carboxamide (dasatinib, BMS-354825) as a potent pan-Src kinase inhibitor. J Med Chem 2006, 49 (23), 6819-6832. DOI: 10.1021/jm060727j ] have been added to the refences’ section
Minor revisions:
Comment 1. Correctly name the genes mentioned throughout the manuscript and the corresponding proteins expressed by them. Italicize the genes and use the correct nomenclature.
Authors ‘response: Thank you very much for your comment. Accordingly, we made the required correction.
Comment 2. Use italics for species names. For example: Drosophila melanogaster, etc.
Authors ‘response: Thank you very much for your comment. Accordingly, we made the required correction.
Comment 3. Standardize the way figures are cited in the text, for example: Figure 1, Figure 2. and if they are cited: ....are outlined as shown (Figure 9), it should be: ...are outlined as shown in Figure 9.
Authors ‘response: Thank you very much for your comment. Accordingly, we made the required correction.
Comment 4. Line 427 refers to Figure 9 when it should be Figure 8.
Authors ‘response: Thank you very much for your comment. Accordingly, we made the required correction.

Reviewer 3 Report
Comments and Suggestions for Authors
Protein kinases are a family of 540 enzymes that transfer a phosphoryl group from a nucleoside triphosphate donor to specific amino acids in target proteins. This family is divided to several subgroups according to the target amino acids and localization, including the ~40 non-receptor protein Tyr kinases (NRTK), which is the subject of the submitted publication. The NRTKs play important roles in the regulation of many cellular processes including proliferation, differentiation, migration, and even apoptosis dependent on the cell type and conditions. These effects are well regulated, and usually occur in response to extracellular stimulations. These stimulations then either activate or inactivate the NRTK by means of autophosphorylation, phosphorylation by other protein kinases or dephosphorylation by various phosphatases. Furthermore, the activity of these kinases is also regulated by a dynamic subcellular localization or interaction with other proteins. Being such central regulatory components, their dysregulation often leads to pathologies including cancer. Therefore, they can serve as therapeutic targets for quite a few diseases.
This review presents all NRTKs describing their secondary structure, and briefly also aspects of their regulation and function in normal as well as cancer cells. In addition, it discusses NRTKs as therapeutic targets for cancer. Although the review aims to be a comprehensive source of knowledge on these kinases, it lacks much information on important aspects of the different NRTKs, including their modes of activation and regulation as well as main roles in distinct cells. In addition, the parts of the writing is slopy and difficult to follow. The following points should be addressed in order to justify publication.
1. It is clear that due to the large number of NRTKs, it is difficult to provide all relevant information on all of them. However, the mode of activation, main substrates and main functions should be mentioned for all NRTKs. Some of this information can be added in tables.
2. It is strongly recommended to add a figure with a three-dimensional conformation of one or two NRTKs, in order to make the description of this issue clearer.
3. In Fig. 2, it is not clear why do the authors show the Ack family kinases from different organisms. This is not done for any other group.
4. In Fig. 5, FRNK is not a kinase, and therefore is not supposed to be included in figure, unless explicitly justified. More information on FRNK should be included in the text.
5. In Fig. 7, the authors show many post-translational modifications, but their nature and function are not properly explained.
6. Fig. 8 is not clear. Why are c-Src and v-Src and CSK included, and CHK is not. Can CSK and CHK be separated into a distinct group. Why the SH4 of c-SRC is not included. Most of the information in the legend should appear in the text, not just the legend. The figure numbering in the text is wrong (Fig. 9).
7. In continuation to point 6, much of the information that appears in the figure is not properly explained in the text. Therefore, it makes it difficult to follow.
8. There are plenty of inaccuracies and mistakes throughout the articles. The article should undergo a stringent proofreading to avoid it. Some examples (out of many) are:
a) In the Simple Summary: Protein tyrosine kinases (PTKs) are classified into two groups, one of which represents tyrosine kinases while the other serine/ threonine kinases. Should be RTK and NRTK.
b) In line 44, NRTKs are kinase enzymes which are overexpressed. The word kinase should be taken out.
c) In line 70, NRTKs are not just cytoplasmic but also nuclear as shown later.
d) The terms SH2 and SH3 are spelled out several times.
e) In the Src section the authors should stick to the term CHK and not Matk.
f) The phosphatases that regulate NRTKs are not mentioned at all.
g) There are several sentences that are very similar to sentences from previous published reviews.
Comments on the Quality of English LanguageThe English is usually OK
Author Response
Authors ‘response to the comment of the reviewer 3
As required enclosed find please our response point-for-point to the valuable comment of the reviewer 03
Comment of the Reviewer 03
Comments and Suggestions for Authors
Protein kinases are a family of 540 enzymes that transfer a phosphoryl group from a nucleoside triphosphate donor to specific amino acids in target proteins. This family is divided to several subgroups according to the target amino acids and localization, including the ~40 non-receptor protein Tyr kinases (NRTK), which is the subject of the submitted publication. The NRTKs play important roles in the regulation of many cellular processes including proliferation, differentiation, migration, and even apoptosis dependent on the cell type and conditions. These effects are well regulated, and usually occur in response to extracellular stimulations. These stimulations then either activate or inactivate the NRTK by means of autophosphorylation, phosphorylation by other protein kinases or dephosphorylation by various phosphatases. Furthermore, the activity of these kinases is also regulated by a dynamic subcellular localization or interaction with other proteins. Being such central regulatory components, their dysregulation often leads to pathologies including cancer. Therefore, they can serve as therapeutic targets for quite a few diseases.
This review presents all NRTKs describing their secondary structure, and briefly also aspects of their regulation and function in normal as well as cancer cells. In addition, it discusses NRTKs as therapeutic targets for cancer. Although the review aims to be a comprehensive source of knowledge on these kinases, it lacks much information on important aspects of the different NRTKs, including their modes of activation and regulation as well as main roles in distinct cells. In addition, the parts of the writing is slopy and difficult to follow. The following points should be addressed in order to justify publication.
Authors ‘response: Thank you very much for your comment. Accordingly, we made the required changes.
Comment1. It is clear that due to the large number of NRTKs, it is difficult to provide all relevant information on all of them. However, the mode of activation, main substrates and main functions should be mentioned for all NRTKs. Some of this information can be added in tables.
Authors ‘response: Thank you very much for your valuable comment. Accordingly, we more information to ABL family kinase. However, the presentation of the mode of activation, and the main substrates and the main function of the different NRTKs in a table was not possible do it according we add the following text to the sections of Ack and ABL kinases
See lines:143-167; the following paragraph [The activation loop conformations of both phosphorylated and unphosphorylated forms of Ack are similar and do not cover the substrate binding site, suggesting that phosphorylation of the activation loop may not play a dramatic stimulatory role [40, 41].
The analysis of the crystal structure and extensive biochemical studies of Ack family member provided insight into its carefully orchestrated process of activation both in normal and cancer cells [27]. The identification of 10 amino acid sequence rich in proline residues within Ack1 upstream of the kinase recognition segment of the MHR revealed an important role for the interaction of the proline rich sequence. With the SH3 to facilitate the orientation of the MHR for the inhibitory interactions of C-terminal region of the kinase domain to stabilize the auto-inhibited state of the Ack1 kinase [ 22, 30, 31].
However, activation of ACK1 results as consequence for the interaction of ligand activated RTKs with the MHR of ACK1 by augmentation attenuating the auto-inhibitory interaction of the MHR with its own kinase domain [22, 31, 42]. While the complete activation of ACK1 has been reported to be attributed to the amino-terminal SAM domain that is essential to facilitate membrane localization and symmetric dimerization to trigger the trans-phosphorylation and activation of Ack1 [22, 27]. To that end, Ack1 is expected to switch to different modes of kinase activation, to be adapted to cellular requirements.
The identification of the substrates of Ack1 have been identified both in normal and cancer cells. For example, the phosphorylation of Wiskott-Aldrich syndrome protein (WASP) by Ack1 has been reported to promote its actin remodeling activity [28 ]. While the phosphorylation of p130Cas, a component of focal adhesion and is a member of the Cas (Crk-associated substrate) family by Ack1 is associated with tumor invasion, promoting cell spreading [ 43] and cell migration [44]. In addition to its role in the regulation of cell adhesion and migration. Ack1 has been also reported to phosphorylate androgen receptor (AR) leading to enhancement of AR-mediated gene transcription [45, 46].
] has been added to the section of Ack Family.
References
See lines: 896-952; the following references:
- Prieto-Echagüe, V.; Gucwa, A.; Craddock, B. P.; Brown, D. A.; Miller, W. T. Cancer-associated mutations activate the nonreceptor tyrosine kinase Ack1. J Biol Chem 2010, 285 (14), 10605-10615. DOI: 10.1074/jbc.M109.060459.
- Hubbard, S. R.; Till, J. H. Protein tyrosine kinase structure and function. Annu Rev Biochem 2000, 69, 373-398. DOI: 10.1146/annurev.biochem.69.1.373.
- Sun, G.; Ayrapetov, M. K. Dissection of the catalytic and regulatory structure-function relationships of Csk protein tyrosine kinase. Front Cell Dev Biol 2023, 11, 1148352. DOI: 10.3389/fcell.2023.1148352.
- Gan, W.; Roux, B. Binding specificity of SH2 domains: insight from free energy simulations. Proteins 2009, 74 (4), 996-1007. DOI: 10.1002/prot.22209.
- Pawson, T.; Gish, G. D.; Nash, P. SH2 domains, interaction modules and cellular wiring. Trends Cell Biol 2001, 11 (12), 504-511. DOI: 10.1016/s0962-8924(01)02154-7.
- Mahajan, K.; Mahajan, N. P. ACK1/TNK2 tyrosine kinase: molecular signaling and evolving role in cancers. Oncogene 2015, 34 (32), 4162-4167. DOI: 10.1038/onc.2014.350.
- Yokoyama, N.; Miller, W. T. Biochemical properties of the Cdc42-associated tyrosine kinase ACK1. Substrate specificity, authphosphorylation, and interaction with Hck. J Biol Chem 2003, 278 (48), 47713-47723. DOI: 10.1074/jbc.M306716200.
- Yang, W.; Cerione, R. A. Cloning and characterization of a novel Cdc42-associated tyrosine kinase, ACK-2, from bovine brain. J Biol Chem 1997, 272 (40), 24819-24824. DOI: 10.1074/jbc.272.40.24819.
- Prieto-Echagüe, V.; Miller, W. T. Regulation of ack-family nonreceptor tyrosine kinases. J Signal Transduct 2011, 2011, 742372. DOI: 10.1155/2011/742372.
- Gajiwala, K. S.; Maegley, K.; Ferre, R.; He, Y. A.; Yu, X. Ack1: activation and regulation by allostery. PLoS One 2013, 8 (1), e53994. DOI: 10.1371/journal.pone.0053994.
- Ahmed, S.; Miller, W. T. The noncatalytic regions of the tyrosine kinase Tnk1 are important for activity and substrate specificity. J Biol Chem 2022, 298 (12), 102664. DOI: 10.1016/j.jbc.2022.102664.
- Umarao, P.; Rath, P. P.; Gourinath, S. Cdc42/Rac Interactive Binding Containing Effector Proteins in Unicellular Protozoans With Reference to Human Host: Locks of the Rho Signaling. Front Genet 2022, 13, 781885. DOI: 10.3389/fgene.2022.781885.
- Pao-Chun, L.; Chan, P. M.; Chan, W.; Manser, E. Cytoplasmic ACK1 interaction with multiple receptor tyrosine kinases is mediated by Grb2: an analysis of ACK1 effects on Axl signaling. J Biol Chem 2009, 284 (50), 34954-34963. DOI: 10.1074/jbc.M109.072660.
- Hayashi, S. Y.; Craddock, B. P.; Miller, W. T. Phosphorylation of Ack1 by the Receptor Tyrosine Kinase Mer. Kinases Phosphatases 2023, 1 (3), 167-180. DOI: 10.3390/kinasesphosphatases1030011.
- Chan, W.; Tian, R.; Lee, Y. F.; Sit, S. T.; Lim, L.; Manser, E. Down-regulation of active ACK1 is mediated by association with the E3 ubiquitin ligase Nedd4-2. J Biol Chem 2009, 284 (12), 8185-8194. DOI: 10.1074/jbc.M806877200.
- Lougheed, J. C.; Chen, R. H.; Mak, P.; Stout, T. J. Crystal structures of the phosphorylated and unphosphorylated kinase domains of the Cdc42-associated tyrosine kinase ACK1. J Biol Chem 2004, 279 (42), 44039-44045. DOI: 10.1074/jbc.M406703200.
- Yang, W.; Lo, C. G.; Dispenza, T.; Cerione, R. A. The Cdc42 target ACK2 directly interacts with clathrin and influences clathrin assembly. J Biol Chem 2001, 276 (20), 17468-17473. DOI: 10.1074/jbc.M010893200.
- Hodder, S.; Fox, M.; Binti Ahmad Mokhtar, A. M.; Mott, H. R.; Owen, D. ACKnowledging the role of the Activated-Cdc42 associated kinase (ACK) in regulating protein stability in cancer. Small GTPases 2023, 14 (1), 14-25. DOI: 10.1080/21541248.2023.2212573.
- [Galisteo, M. L.; Yang, Y.; Ureña, J.; Schlessinger, J. Activation of the nonreceptor protein tyrosine kinase Ack by multiple extracellular stimuli. Proc Natl Acad Sci U S A 2006, 103 (26), 9796-9801. DOI: 10.1073/pnas.0603714103.
- Linseman, D. A.; Heidenreich, K. A.; Fisher, S. K. Stimulation of M3 muscarinic receptors induces phosphorylation of the Cdc42 effector activated Cdc42Hs-associated kinase-1 via a Fyn tyrosine kinase signaling pathway. J Biol Chem 2001, 276 (8), 5622-5628. DOI: 10.1074/jbc.M006812200
- La Torre, A.; del Mar Masdeu, M.; Cotrufo, T.; Moubarak, R. S.; del Río, J. A.; Comella, J. X.; Soriano, E.; Ureña, J. M. A role for the tyrosine kinase ACK1 in neurotrophin signaling and neuronal extension and branching. Cell Death Dis 2013, 4 (4), e602. DOI: 10.1038/cddis.2013.99.
- Mahajan, N. P.; Whang, Y. E.; Mohler, J. L.; Earp, H. S. Activated tyrosine kinase Ack1 promotes prostate tumorigenesis: role of Ack1 in polyubiquitination of tumor suppressor Wwox. Cancer Res 2005, 65 (22), 10514-10523. DOI: 10.1158/0008-5472.CAN-05-1127.
- van der Horst, E. H.; Degenhardt, Y. Y.; Strelow, A.; Slavin, A.; Chinn, L.; Orf, J.; Rong, M.; Li, S.; See, L. H.; Nguyen, K. Q.; et al. Metastatic properties and genomic amplification of the tyrosine kinase gene ACK1. Proc Natl Acad Sci U S A 2005, 102 (44), 15901-15906. DOI: 10.1073/pnas.0508014102.
- Kim, E. H.; Cao, D.; Mahajan, N. P.; Andriole, G. L.; Mahajan, K. ACK1-AR and AR-HOXB13 signaling axes: epigenetic regulation of lethal prostate cancers. NAR Cancer 2020, 2 (3), zcaa018. DOI: 10.1093/narcan/zcaa018.
- Mahajan, K.; Coppola, D.; Challa, S.; Fang, B.; Chen, Y. A.; Zhu, W.; Lopez, A. S.; Koomen, J.; Engelman, R. W.; Rivera, C.; et al. Ack1 mediated AKT/PKB tyrosine 176 phosphorylation regulates its activation. PLoS One 2010, 5 (3), e9646. DOI: 10.1371/journal.pone.0009646.] have been added to the section of references
See lines: 484-589; the following Paragraph [The activation of ABL family members (e.g. Abl1, ARG) by different mechanisms including, mutation, dimerization, phosphorylation, or binding proteins that disrupt intramolecular interactions [200-202], and by the phosphorylation within kinase and interlinked regions (Y412, Y245) [203-206]. However, the disruption of autoinhibition in response to the translocation of Abl1 or Abl2 next to a variety of different genes (e.g., BCR, Tel, ETV6), causes constitutive activation leading to the development different tumor types including leukemia [207]. Although both c-Abl and Arg are involved in the development of a variety of human leukemias, mutations and/or activating translocations of these kinases so far are not identified in solid tumors [208]. Therefore, it is expected that c-Abl and Arg are not activated in other cancers. However, data from the immune histochemistry analysis of c-Abl and/or Arg showed that they are overexpressed in some solid tumors, including brain, lung, ovarian, colon and prostate cancer, as well as in chondrosarcomas, liposarcomas, diffuse gastric adenocarcinomas, oral squamous cell carcinomas, atypical teratoid and malignant rhabdoid tumors and endometrial carcinomas, compared to normal tissue or benign tumors. In addition, c-Abl amplification was found in renal medulla carcinomas [209-211]. The structure and functional domains, as well as the 3D- of ABL kinase protein are outlined in figure 9 A and B. ]to the section of ABL family
References
See lines : 1315-1343 ; The following references :
- Aoyama, K.; Yuki, R.; Horiike, Y.; Kubota, S.; Yamaguchi, N.; Morii, M.; Ishibashi, K.; Nakayama, Y.; Kuga, T.; Hashimoto, Y.; et al. Formation of long and winding nuclear F-actin bundles by nuclear c-Abl tyrosine kinase. Exp Cell Res 2013, 319 (20), 3251-3268. DOI: 10.1016/j.yexcr.2013.09.003.
- Hu, H.; Bliss, J. M.; Wang, Y.; Colicelli, J. RIN1 is an ABL tyrosine kinase activator and a regulator of epithelial-cell adhesion and migration. Curr Biol 2005, 15 (9), 815-823. DOI: 10.1016/j.cub.2005.03.049.
- Sriram, G.; Reichman, C.; Tunceroglu, A.; Kaushal, N.; Saleh, T.; Machida, K.; Mayer, B.; Ge, Q.; Li, J.; Hornbeck, P.; et al. Phosphorylation of Crk on tyrosine 251 in the RT loop of the SH3C domain promotes Abl kinase transactivation. Oncogene 2011, 30 (46), 4645-4655. DOI: 10.1038/onc.2011.170.
- Cao, X.; Tanis, K. Q.; Koleske, A. J.; Colicelli, J. Enhancement of ABL kinase catalytic efficiency by a direct binding regulator is independent of other regulatory mechanisms. J Biol Chem 2008, 283 (46), 31401-31407. DOI: 10.1074/jbc.M804002200.
- Plattner, R.; Kadlec, L.; DeMali, K. A.; Kazlauskas, A.; Pendergast, A. M. c-Abl is activated by growth factors and Src family kinases and has a role in the cellular response to PDGF. Genes Dev 1999, 13 (18), 2400-2411. DOI: 10.1101/gad.13.18.2400.
- Brasher, B. B.; Van Etten, R. A. c-Abl has high intrinsic tyrosine kinase activity that is stimulated by mutation of the Src homology 3 domain and by autophosphorylation at two distinct regulatory tyrosines. J Biol Chem 2000, 275 (45), 35631-35637. DOI: 10.1074/jbc.M005401200.
- Smith, K. M.; Van Etten, R. A. Activation of c-Abl kinase activity and transformation by a chemical inducer of dimerization. J Biol Chem 2001, 276 (26), 24372-24379. DOI: 10.1074/jbc.M100786200.
- Fan, P. D.; Cong, F.; Goff, S. P. Homo- and hetero-oligomerization of the c-Abl kinase and Abelson-interactor-1. Cancer Res 2003, 63 (4), 873-877
- Sawyers, C. L.; McLaughlin, J.; Goga, A.; Havlik, M.; Witte, O. The nuclear tyrosine kinase c-Abl negatively regulates cell growth. Cell 1994, 77 (1), 121-131. DOI: 10.1016/0092-8674(94)90240-2.
- Greuber, E. K.; Smith-Pearson, P.; Wang, J.; Pendergast, A. M. Role of ABL family kinases in cancer: from leukaemia to solid tumours. Nat Rev Cancer 2013, 13 (8), 559-571. DOI: 10.1038/nrc3563.
- Ganguly, S. S.; Fiore, L. S.; Sims, J. T.; Friend, J. W.; Srinivasan, D.; Thacker, M. A.; Cibull, M. L.; Wang, C.; Novak, M.; Kaetzel, D. M.; et al. c-Abl and Arg are activated in human primary melanomas, promote melanoma cell invasion via distinct pathways, and drive metastatic progression. Oncogene 2012, 31 (14), 1804-1816. DOI: 10.1038/onc.2011.361.
- Koos, B.; Jeibmann, A.; Lünenbürger, H.; Mertsch, S.; Nupponen, N. N.; Roselli, A.; Leuschner, I.; Paulus, W.; Frühwald, M. C.; Hasselblatt, M. The tyrosine kinase c-Abl promotes proliferation and is expressed in atypical teratoid and malignant rhabdoid tumors. Cancer 2010, 116 (21), 5075-5081. DOI: 10.1002/cncr.25420. ] have been added to references section
Comment 2. It is strongly recommended to add a figure with a three-dimensional conformation of one or two NRTKs, in order to make the description of this issue clearer.
Authors ‘response: Thank you very much for your comment. Accordingly, we demonstrated the three-dimensional confirmation of ABL kinase.
The following changes have been made
Figures
See Figure 9B; The following figure: Figure 9B, has been created
Legends to Figures
See Lines: 513-515; the following legend [Figure 9B. Three-dimensional conformation of ABL kinase protein. The different domains of the ABL kinase including SH3 Green), SH2 (red), and SH1/kinase (yellow) domains, and the catalytic pocket, allosteric pocket, activation loop and the carboxy-terminal helix are demonstrated ] has been added to the main Text of the manuscript
Comment 3. In Fig. 2, it is not clear why do the authors show the Ack family kinases from different organisms. This is not done for any other group.
Authors ‘response: Thank you very much for your valuable comment. Accordingly, we removed the diagrams of the other organisms
The following changes have been made in Fig. 2.
See Fig.2, the diagrams of the other organisms have been removed.
Comment 4. In Fig. 5, FRNK is not a kinase, and therefore is not supposed to be included in figure, unless explicitly justified. More information on FRNK should be included in the text.
Authors ‘response: Thank you very much for your comment. As required, we removed FRNK from the diagram and modified legend to fig. 5.
Comment 5. In Fig. 7, the authors show many post-translational modifications, but their nature and function are not properly explained.
Authors ‘response: Thank you very much for your comment. Accordingly, we added information about the post-transcriptional modification.
The following modifications have been made
See lines: 444-448 ; the following paragraph [ SFKs are highly homologous in structure that consists of four consecutive Src homology (SH4, HS3, SH2 and SH1) domains [168, 174 ]. Of note, the SH4 domain serves as a membrane targeting region for myristylation and/or palmitoylation at the N-terminus of SFK members [175 ]. Thus, the myristylation and/or palmitoylation of SH4 is essential to facilitate the membrane localization of SKF members [ 176]. ] has been added
References
See Lines:1253-1259 ; the following references:
- McClendon, C. J.; Miller, W. T. Structure, Function, and Regulation of the SRMS Tyrosine Kinase. Int J Mol Sci 2020, 21 (12). DOI: 10.3390/ijms21124233.
- Liang, X.; Lu, Y.; Wilkes, M.; Neubert, T. A.; Resh, M. D. The N-terminal SH4 region of the Src family kinase Fyn is modified by methylation and heterogeneous fatty acylation: role in membrane targeting, cell adhesion, and spreading. J Biol Chem 2004, 279 (9), 8133-8139. DOI: 10.1074/jbc.M311180200.
- Fhu, C. W.; Ali, A. Protein Lipidation by Palmitoylation and Myristoylation in Cancer. Front Cell Dev Biol 2021, 9, 673647. DOI: 10.3389/fcell.2021.673647.] have been added the references ‘ section
Comment 6. Fig. 8 is not clear. Why are c-Src and v-Src and CSK included, and CHK is not. Can CSK and CHK be separated into a distinct group. Why the SH4 of c-SRC is not included. Most of the information in the legend should appear in the text, not just the legend. The figure numbering in the text is wrong (Fig. 9). In continuation to point 6, much of the information that appears in the figure is not properly explained in the text. Therefore, it makes it difficult to follow
Authors ‘response: Thank you very much for your comment. As required modified the figure and the text as require.
Comment 7. There are plenty of inaccuracies and mistakes throughout the articles. The article should undergo a stringent proofreading to avoid it. Some examples (out of many) are:
- a) In the Simple Summary: Protein tyrosine kinases (PTKs) are classified into two groups, one of which represents tyrosine kinases while the other serine/ threonine kinases. Should be RTK and NRTK.
Authors ‘response: Thank you very much for your comment. Accordingly, we corrected the mistakes overall in the main text of the manuscript
- b) In line 44, NRTKs are kinase enzymes which are overexpressed. The word kinase should be taken out.
Authors ‘response: Thank you very much for your comment. We made the required correction and removed the word kinase
- c) In line 70, NRTKs are not just cytoplasmic but also nuclear as shown later.
Authors ‘response: Thank you very much for your valuable comment. Accordingly, we modified the text to be more suitable.
See lines: 73-76; the following text [ NRTKs are both cytoplasmic and nuclear proteins, the localization of the cytoplasmic NRTK proteins are either free or anchored to the inner side of the cell membrane [15, 16]. The main function of the cytoplasmic NRTK protein is to mediate intracellular signals resulting from extracellular receptors-dependent activation [15, 17]] has been added.
- d) The terms SH2 and SH3 are spelled out several times.
Authors ‘response: Thank you very much for your commen. Accordingly, we made the required correction.
- e) In the Src section the authors should stick to the term CHK and not Matk.
Authors ‘response: Thank you very much for your comment. Accordingly, we made the required Modification of the text.
See lines: 393, 396, 401 and 403
- f) The phosphatases that regulate NRTKs are not mentioned at all.
Authors ‘response: Thank you very much. We added some of these phosphatases to the text.
Main Text
See lines: 796-814 ; the following paragraph [5. Non-receptor tyrosine kinases and protein tyrosine phosphatases
NRTKs are regulated by protein tyrosine phosphatases (PTPs), which dephosphorylate protein targets and balance the activity of protein kinases [11, 351] . PTPs play a crucial role in the regulation of tyrosine phosphorylation and signal transduction [352, 353].
These PTPs carry a functionally conserved PTPase domain with a highly conserved signature motif (His/Val)Cys(X)5Arg(Ser/Thr), which utilizes the catalytic function to hydrolyze phosphate from the tyro-sine residues of their substrates [5 ]. PTPs are classified as tyrosine-specific phosphatase or dual-specific phosphatases [354, 355]. Dual-specific PTPs are characterized by their ability to dephosphorylate both tyrosine and serine/threonine residues [354, 355]. However, based cellular localization PTPs can also classified into receptor or non-receptor types PTPs receptor types referred to PTPs that span the cell membrane, while the non-receptor PTPs referred to PTPs that are localized mainly in the cytoplasm [356, 357]. In addition to their above-mentioned classifications, PTPs can be classified into tumor suppressive or oncogenic PTPs based on the type of dephosphorylated protein kinase of interest [358, 359]. Src homology-2-containing protein tyrosine phosphatase 2 (SHP-2) is a non-receptor, tyrosine-specific, PTP that binds to the intracellular domain of RTK via phosphotyrosine residues [360, 361]. SHP-2 can dephosphorylate proteins or residues that are able to suppress the function of Ras and SFK inhibitors to enhance the activity of tumorigenic pathways such as PI3K and MAPK/ERK signaling [362, 363 ].
] has been added to the text
References
See lines: 1673-1701; The following references:
- Li, H.; Zhang, P.; Liu, C.; Wang, Y.; Deng, Y.; Dong, W.; Yu, Y. The Structure, Function and Regulation of Protein Tyrosine Phosphatase Receptor Type J and Its Role in Diseases. Cells 2022, 12(1). DOI: 10.3390/cells12010008.
- Hale, A. J.; Ter Steege, E.; den Hertog, J. Recent advances in understanding the role of protein-tyrosine phosphatases in development and disease. Dev Biol 2017, 428(2), 283-292. DOI: 10.1016/j.ydbio.2017.03.023.
- Xie, F.; Dong, H.; Zhang, H. Regulatory Functions of Protein Tyrosine Phosphatase Receptor Type O in Immune Cells. Front Immunol 2021, 12, 783370. DOI: 10.3389/fimmu.2021.783370.
- Bollu, L. R.; Mazumdar, A.; Savage, M. I.; Brown, P. H. Molecular Pathways: Targeting Protein Tyrosine Phosphatases in Cancer. Clin Cancer Res 2017, 23(9), 2136-2142. DOI: 10.1158/1078-0432.CCR-16-0934.
- Motiwala, T.; Jacob, S. T. Role of protein tyrosine phosphatases in cancer. Prog Nucleic Acid Res Mol Biol 2006, 81, 297-329. DOI: 10.1016/S0079-6603(06)81008-1.
- Motiwala, T.; Jacob, S. T. Role of protein tyrosine phosphatases in cancer. Prog Nucleic Acid Res Mol Biol 2006, 81, 297-329. DOI: 10.1016/S0079-6603(06)81008-1.
- Gao, P. P.; Qi, X. W.; Sun, N.; Sun, Y. Y.; Zhang, Y.; Tan, X. N.; Ding, J.; Han, F. The emerging roles of dual-specificity phosphatases and their specific characteristics in human cancer. Biochim Biophys Acta Rev Cancer 2021, 1876(1), 188562. DOI: 10.1016/j.bbcan.2021.18856.
- Sivaganesh, V.; Scanlon, C.; Iskander, A.; Maher, S.; Lê, T.; Peethambaran, B. Protein Tyrosine Phosphatases: Mechanisms in Cancer. Int J Mol Sci 2021, 22(23). DOI: 10.3390/ijms222312865.
- Du, Y.; Grandis, J. R. Receptor-type protein tyrosine phosphatases in cancer. Chin J Cancer 2015, 34(2), 61-69. DOI: 10.5732/cjc.014.10146.
- Osman, N.; Lucas, S.; Cantrell, D. The role of tyrosine phosphorylation in the interaction of cellular tyrosine kinases with the T cell receptor zeta chain tyrosine-based activation motif. Eur J Immunol 1995, 25(10), 2863-2869. DOI: 10.1002/eji.1830251023.
- Stein-Gerlach, M.; Wallasch, C.; Ullrich, A. SHP-2, SH2-containing protein tyrosine phosphatase-2. Int J Biochem Cell Biol 1998, 30(5), 559-566. DOI: 10.1016/s1357-2725(98)00002-8.; [361 ] Jadwin, J. A.; Curran, T. G.; Lafontaine, A. T.; White, F. M.; Mayer, B. J. Src homology 2 domains enhance tyrosine phosphorylation. J Biol Chem 2018, 293 (2), 623-637. DOI: 10.1074/jbc.M117.794412.
- Song, Y.; Zhao, M.; Zhang, H.; Yu, B. Double-edged roles of protein tyrosine phosphatase SHP2 in cancer and its inhibitors in clinical trials. Pharmacol Ther 2022, 230, 107966. DOI: 10.1016/j.pharmthera.2021.107966.
- Matozaki, T.; Murata, Y.; Saito, Y.; Okazawa, H.; Ohnishi, H. Protein tyrosine phosphatase SHP-2: a proto-oncogene product that promotes Ras activation. Cancer Sci 2009, 100 (10), 1786-1793. DOI: 10.1111/j.1349-7006.2009.01257.x. . ] have been added to the section of the references
- g) There are several sentences that are very similar to sentences from previous published reviews.
Authors ‘response: Thank you very much for your comment. Accordingly, we modified the text as required, whenever its possible
Comments on the Quality of English Language
The English is usually OK
Authors ‘response: Thank you for your comment

Round 2
Reviewer 2 Report
Comments and Suggestions for Authors
The authors have made a great effort to take into account all my observations and suggestions in order to produce a better version of this manuscript.
Reviewer 3 Report
Comments and Suggestions for Authors The paper should be published in its present form and I have no further comments. Comments on the Quality of English Language The paper should be published in its present form and I have no further comments.